
1     **Emission of volatile halogenated organic compounds**

2     **over various landforms at the Dead Sea**

Moshe Shechner[1] Alex Guenther[2], Robert Rhew[3], Asher Wishkerman[4], Qian Li[1], Donald Blake[2], Gil Lerner[1] and
Eran Tas[1]*
[1]The Robert H. Smith Faculty of Agricultural, Food & Environment, Department of Soil and Water Sciences, The
Hebrew University of Jerusalem, Rehovot, Israel
[2]Department of Earth System Science, University of California, Irvine, CA, USA;  Department of Geography at
Berkeley
[3] Department of Geography and Berkeley Atmospheric Sciences Center, University of California, Berkeley,
Berkeley, California 94720, United States
[4]Ruppin Academic Center, Michmoret, Israel; Department of Chemistry, University of California, Irvine, Irvine,
CA 92697
* Corresponding author –Eran Tas, The Department of Soil and Water Sciences, The Robert H.
Smith Faculty of Agriculture, Food and Environment, Hebrew University of Jerusalem,
Rehovot, Israel. eran.tas@mail.huji.ac.il.





**Abstract.** Volatile halogenated organic compounds (VHOCs), such as methyl halides ($CH_3X$;
X=Br, Cl and I) and very short-lived halogenated substances (VSLS; $CHBr_3$, $CH_2Br_2$, $CHBrCl_2$,
$C_2HCl_3$, $CHCl_3$ and $CHBr_2Cl$) are well known for their significant influence on ozone
concentrations and oxidation capacity of the troposphere and stratosphere, and for their key role
in aerosol formation. Insufficient characterization of the sources and emission rate of VHOCs
limits our present ability to understand and assess their impact in both the troposphere and the
stratosphere. Over the last two decades several natural terrestrial sources for VHOCs, including
soil and vegetation, have been identified, but our knowledge about emission rates from these
sources and their responses to changes in ambient conditions remains limited. Here we report
measurements of the mixing ratios and the fluxes of several chlorinated and brominated VHOCs
from different landforms and vegetated sites at the Dead Sea during different seasons. Fluxes
were highly variable but were generally positive (emissive), corresponding with elevated mixing
ratios for all of the VHOCs investigated in the four investigated site types — bare soil, coastal,
cultivated and natural vegetated sites — except for fluxes of $CH_3I$ and $C_2HCl_3$ over the
vegetated sites. In contrast to previous reports, we also observed emissions of brominated
trihalomethanes, with net molar fluxes ordered as follows: $CHBr_2Cl$ > $CHBr_3$ > $CHBrCl_2$ >
$CHCl_3$. This finding can be explained by the enrichment of soil with Br. Correlation analysis, in
agreement with recent studies, indicated common controls for the formation and emission of all
the above trihalomethanes but also for $CH_2Br_2$. Also in line with previous reports, we observed
elevated emissions of $CHCl_3$ and $C_2HCl_3$ from mixtures of soil and different salt-deposited
structures; the high correlations of flux with methyl halides, and particularly with $CH_3I$,
suggested that at least $CH_3I$ is also emitted via similar mechanisms or is subjected to similar
controls. Overall, our results indicate elevate emission of VHOCs from bare soil under semi-arid
conditions. Along with other recent studies, our findings point to the strong emission potential
of a suite of VHOCs from saline soils and salt lakes, and call for additional studies of emission
rates and mechanisms of VHOCs from saline soils and salt lakes.





## 1 Introduction

Volatile halogenated organic compounds (VHOCs), such as methyl halides ($CH_3X$; X=Br, Cl and I) and very short-lived halogenated substances (VSLS) contribute substantially to the loading of tropospheric and lower stratospheric reactive halogen species (RHS, containing Cl, Br or I and their oxides) (Carpenter and Reimann et al., 2014;Carpenter et al., 2013;Derendorp et al., 2012). RHS in turn lead to destruction of ozone ($O_3$), changes in atmospheric oxidation capacity, and radiative forcing (Simpson et al., 2015). Depletion of $O_3$ in the stratosphere is associated with damage to biological tissues owing to an increase in transmittance of UVB radiation (Rousseaux et al., 1999). In the troposphere $O_3$ depletion is of great importance, given that $O_3$ is toxic to humans, plants, and animals, is a greenhouse gas, and plays a key role in the oxidation capacity of the atmosphere.

Owing to their relatively short lifetimes (<6 months) the transport of VSLS to the stratosphere occurs primarily in the tropics, where deep convection is frequent. Brominated VSLS primarily originate from the ocean whereas chlorinated VSLS, except for $CHCl_3$ and $C_2H_5Cl$, originate primarily from anthropogenic sources. $CH_3I$, having a relatively short lifetime, is also classified as a VSLS, and contributes significantly to tropospheric $O_3$ destruction in the marine boundary layer (MBL) (Carpenter and Reimann et al., 2014) and also, indirectly, to cloud condensation nuclei formation (O'Dowd et al., 2002). It is now well established that emission of brominated (e.g., $CHBr_3$, $CH_2Br_2$, and $CHClBr_2$) and iodinated (e.g., $CH_3I$) VSLS tends to be much larger in coastal areas than in the open ocean (Carpenter et al., 2009;Carpenter et al., 2000;Liu et al., 2011;Bondu et al., 2008;Manley and Dastoor, 1988;Quack and Wallace, 2004), since in the former they can also be emitted from macroalgae under oxidative stress at low tide (Pedersen et al., 1996). The ocean is also a major source of $CH_3Br$, and a significant (~19 %) source of $CH_3Cl$ (Carpenter and Reimann et al., 2014), as they originate from phytoplankton, bacteria, and detritus.



Despite the numerous efforts made in recent years to evaluate halocarbon budgets,
uncertainties still exist concerning the strengths of both their sources and their sinks. The
budgets of $CH_3Br$ and $CH_3Cl$ are unbalanced, with sinks outweighing sources by ~32 % and
~17 %, respectively (Carpenter and Reimann et al., 2014). Uncertainties in the global budgets
of naturally occurring VSLS are large, with discrepancies having a factor of ~2−3 between top-
down and bottom-up emission inventories (Carpenter and Reimann et al., 2014). This results
largely from poor characterisation of emission sources (Warwick et al., 2006;Hossaini et al.,
2013;Ziska et al., 2013).

Studies over the past few decades have clearly demonstrated that terrestrial sources also

constitute a major fraction of the atmospheric budget for both methyl halides and VSLS
(Carpenter and Reimann et al., 2014). Many terrestrial plants have been identified as sources of
$CH_3Cl$ (Yokouchi et al., 2007), and the results of recent modelling indicate that about 55 % of
the global sources of $CH_3Cl$ originate from tropical lands (Xiao et al., 2010;Carpenter and
Reimann et al., 2014). It was also suggested that natural terrestrial sources of $CH_3Br$, especially
emissions from terrestrial vegetation, must account for a large part of the missing sources
(Gebhardt et al., 2008;Yassaa et al., 2009;Warwick et al., 2006;Gan et al., 1998;Yokouchi et al.,
2002;Moore, 2006;Rhew et al., 2001;Wishkerman et al., 2008), and emissions have been
observed from peatlands, wetlands, salt marshes, shrublands, forests, and some cultivated crops
(Gan et al., 1998;Varner et al., 1999;Lee-Taylor and Holland, 2000). $CHCl_3$ was also found to
be emitted from various terrestrial sources, including rice, soil, tundra, forest floor, and different
types of microorganisms such as fungi and termites (see Dimmer et al. (2001) and (Rhew et al.,

2008)).

The importance of VHOC emission from soil, sediments, and salt lake deposits was recently
recognized (see Kotte et al. (2012), Ruecker et al. (2014), and references therein). For example,
Keppler et al. (2000) revealed natural abiotic emission of $CH_3Br$, $CH_3Cl$, and $CH_3I$ as well as
additional chlorinated VHOCs from soil and sediments harboring an oxidant such as Fe(III),



halides, and organic matter (OM), while Weissflog et al. (2005) found that salt lake sediments
can be a source for several C1 and C2 chlorinated species, including $CHCl_3$ and $C_2HCl_3$,
induced by halobacteria in the presence of dissolved Fe. Huber et al. (2009) identified an abiotic
natural emission of trihalomethanes from soil, including $CHCl_3$, $CHBrCl_2$, and $CHBr_2Cl$,
induced by oxidation of OM by Fe(III) and hydrogen peroxide, while Hoekstra et al. (1998)
identified natural emission of $CHBr_3$ following enrichment of the soil by KBr. In addition,
Carpenter et al. (2005) identified $CHBr_3$ emission from a peatland or another terrestrial source at
Mace Head. Albers et al. (2017) revealed that $CHCl_3$, $CHBrCl_2$, and potentially also other
trihalomethanes can be emitted from soils, probably induced by hydrolysis of trihaloacetyl
compounds. Several other studies report strong emissions of $CH_3Cl$, $CH_3Br$, and $CH_3I$ from
coastal marsh vegetation and to a lesser extent from the marsh's soil (Rhew et al., 2002;Rhew et
al., 2001;Rhew et al., 2014;Wishkerman et al., 2008;Rhew et al., 2000), with significant
importance on a global scale (Deventer et al., 2018;Manley et al., 2006). In addition, peatland
has been indicated as an important source for $CH_3Br$, $CH_3Cl$, $CH_3I$ and $CHCl_3$ (Simmonds et al.,
2010;Khan et al., 2012;Dimmer et al., 2001;Carpenter et al., 2005), and Sive et al. (2007)
identified a globally significant source of $CH_3I$ from mid-latitude vegetation and soil.

Accordingly, the need for improved understanding of VHOC emission from saline

environments and their potential importance on the global scale have been highlighted by recent
studies (Weissflog et al., 2005;Kotte et al., 2012;Ruecker et al., 2014;Deventer et al., 2018).
Moreover, owing to global warming, saline environments are likely to become more prevalent
(IPCC 2007;Ruecker et al., 2014). The present study is aimed at improving our knowledge about
the emission of VHOCs from salt lake environments by quantifying the flux and the mixing
ratios of methyl halides and halogenated VSLS from different sites in the area of the Dead Sea.

The Dead Sea is unique because it is the lowest point on the Earth's surface, about 430 m

below sea level, with water salinity and $[Br^-]/[Cl^-]$ ratio 12 and 7.5 times higher than in normal
ocean waters, respectively. Fast evaporation from the sea leads to a variety of newly exposed



landforms. Despite the high salinity, emission of VHOCs via biotic processes at the Dead Sea is

also potentially feasible. The unicellular green alga *Dunaliella parva* was found to be active in

Dead Sea water (Oren and Shilo, 1985), while additional bacteria and fungi that were isolated

from the sea could also potentially be active under the extreme conditions (Oren et al.,

2008;Jacob et al., 2017;Buchalo et al., 1998). Mycobiota, including fungi and biota, were also

detected in the Dead Sea's hypersaline soil and coastal sand (Pen-Mouratov et al., 2010;Kis-

Papo et al., 2001;Jacob et al., 2017).

Studying the emission of VHOCs at the Dead Sea is also interesting, in view of local

sharp ozone depletion events (Hebestreit et al., 1999;Tas et al., 2003;Matveev et al.,

2001;Zingler and Platt, 2005;Tas et al., 2006) as well as mercury depletion events (Tas et al.,

2012;Obrist et al., 2011) in the boundary layer at this area. Emissions of brominated and

iodinated VSLS can potentially lead to formation of the reactive iodine and bromine species that

are responsible for these processes.

## 2 Methods

### 2.1 Field measurements and samplings

Field measurements were taken at selected sites along the Dead Sea to measure the mixing

ratios and evaluate the vertical flux of VHOCs over different land-use types, seasons, and

distance from the seawater, as summarized in Table 1. Soil samples from the various sites were

analyzed and meteorological measurements were performed in situ, as described below.

### 2.1.1 Measurement sites

All measurements were taken at the Dead Sea area. The Dead Sea's geographic position is

between 31°50' N and 31°00' N, 35°30' E, about 430 m below sea level. It is located in a semi-

arid area with a very high seawater evaporation rate of 400 cm y$^{-1}$ (Alpert et al., 1997), and has

only a low rate of freshwater inflow. As a result, the water salinity is 12 times higher than that





of normal ocean water. Dead Sea water contains on average 5.6 g L$^{-1}$ bromide and 225 g L$^{-1}$
chloride (Br$^{-}$/Cl$^{-}$ ratio ≈ 0.025) (Niemi, 1997), whereas normal ocean water contains 0.065 g L$^{-1}$
bromide and 19 g L$^{-1}$ chloride (Br$^{-}$/Cl$^{-}$ ratio ≈ 0.0034) (Sverdrup, 1942).

All measurement sites are nearly flat and are located either along or near the Dead Sea coast

(see Fig. 1). Overall, for our investigations we selected emissions from bare soil sites (BARE) at
Mishmar (MSMR; BARE−MSMR) and at Massada (MSD; BARE−MSD), coastal sites that are
mixtures of soil and salt deposits (COAST) at Ein-Gedi (EGD; COAST−EGD) and Tzukim
(TKM; COAST−TKM), natural Tamarix vegetation at Ein Tamar (ET; TMRX−ET), cultivated
watermelon agricultural field at Kalya (KLY; WM−KLY), and directly from the seawater at
Kedem (KDM; SEA−KDM). Note that at SEA−KDM we did not evaluate fluxes. Based on in-
situ wind direction measurements, the sampled air masses at SEA−KDM were transported over
the seawater from the east (see Fig. 1), at least 1 h prior to sampling and during the sampling. To
study the effect of distance from the seawater on emission rates, measurements at both
COAST−EGD and COAST−TKM were taken at three and two different distances from the sea,
respectively. The shorter, middle, and longer distances from the seawater are termed,
respectively, SD, MD and LD. Emission rates at both COAST−EGD and COAST−TKM could
potentially be affected by the distance from the seashore; there are several reasons for this,
including changes across the sites in salt and water soil content and changes in density of the
extremely sparse vegetation cover. In addition, depending on the local wind direction at
COAST−TKM-SD and COAST−EGD-SD, direct emission and uptake from the seawater can
potentially affect the samplings.

In the following we briefly describe the different measurement sites, while additional

information about the sites and measurements is provided in Table 1. BARE−MSMR has a bare
soil consisting of loess and a small fraction of drifted soil covered with small stones and
extremely sparse vegetation, and is located in a valley 1.5 km to the west of the Dead Sea shore.
MSD has bare Hamada soil, with small stones and loess, and is located 2.1 km to the west of the





Dead Sea. COAST−EGD-SD has a dried-out bare saline soil, mixed with salty beds and rocks
and obtaining a small contribution of fresh water inflow at the Dead Sea shore. COAST−EGD-
MD has a dried-out sea bed of bare saline soil, mixed with salty beds and rocks, 0.3 km west of
the Dead Sea shore. COAST−EGD-LD is a dried-out sea bed of loess saline bare soil, mixed
with drifted soil, 0.8 km from the Dead Sea shore. COAST−TKM-SD is a wetted bare soil with
salt deposits, groundwater inflow from the Dead Sea, and minor (<5 %) fresh water inflow lines
covered with perennial grasses found in wetlands (e.g., *Phragmites* sp.), about 0.5 km from the
shore. COAST−TKM-LD is a flat rocky loess area about 1.5 km from the shore, with patchy
salts and sparse mixed shallow vegetation including mostly small *Atriplex* sp., *Tamarix* sp. and
*Retama raetam*. TMRX−ET is a moderately dense *Tamarix* shrubland, with sandy soil, located
1.7 km south of the southern tip of the Dead Sea evaporation ponds. WM−KLY is a well-
irrigated flat cultivated watermelon agricultural field located 2.5 km NW of the Dead Sea shore.

**2.1.2 Field measurements and sampled air analysis**

Air was sampled at each site by placing three different canisters at specified heights (see Table
1) along a meteorological tower. The samples were used to quantify the mixing ratios of
different VHOCs in the air, and their corresponding fluxes calculated by applying the flux-
gradient method (see (Stull 1988;Maier and Schack-Kirchner, 2014;Meredith et al., 2014)). All
canisters were placed high enough above the ground to ensure that all samplings were
performed within the inertial sublayer, except for the lowest canister at TMRX−ET. To
minimize non-synchronized air sampling by the three canisters, we constructed a special system
that allows a fast and almost simultaneous lifting of the canisters. Facilitated by passive grab
samplers (RESTEK Corporation, PA, U.S.), we performed each sampling within 20 minutes by
pulling air into evacuated 1.9 L stainless steel canisters, resulting in an internal canister pressure
higher than 600 torr. Meteorological parameters, including temperature and relative humidity,
wind speed and direction, and global solar radiation, were all continuously measured, starting at





least 30 min before air sampling was initiated. All canisters were sent to the Blake/Rowland
group, University of California, Irvine (UCI), where they were subjected to the analytical
techniques described in detail in Colman et al. (2001). Analyses were performed using gas
chromatography combined with mass spectrometry, flame ionization detection and electron
capture detection to quantify the air mixing ratios of bromoform ($CHBr_3$), trichloroethene
($C_2HCl_3$), methylene bromide ($CH_2Br_2$), dibromochloromethane ($CHBr_2Cl$),
bromodichloromethane ($CHBrCl_2$), trichloroethene ($C_2HCl_3$), chloroform ($CHCl_3$), methyl
iodide ($CH_3I$), methyl bromide ($CH_3Br$) and methyl chloride ($CH_3Cl$). For all gases, accuracy
ranged between 1 % and 10 % and analytical precision between 1 % and 5 % (see Table S1).
Note that the mid-height canister analysis of TMRX−ET-1 indicated a mixing ratio for $CH_3Cl$
that seemed not to agree with any other measured mixing ratios for this species. We therefore
excluded this measurement from all our calculations and used only the lowest and the highest
canisters in the flux calculation for TMRX−ET−1, which may reflect less accurate flux
evaluation. This potentially less accurate flux evaluation is indicated in all relevant figures and
tables.



**Table 1.** Summary of volatile halogenated organic compounds over the Dead Sea. The table records the date, time,
site name (and abbreviation), sampling height, and whether the sampling could potentially be influenced by
emission from the seawater and by precipitation prior to sampling.

| Date dd/m/yyyy | Time (Local) | Site name / measurement abbreviation [a] | Sampling heights (m) | Seawater [b] | Precipitation (days before sampling) [c] |
|---|---|---|---|---|---|
| 20/4/2016 | 08:45−08:55 | BARE−MSMR / BARE−MSMR-1 | 2.5, 4.5, 7.0 | − | > 3 months |
| 21/4/2016 | 08:45−08:55 | WM−KLY / WM−KLY-1 | 1.0,2.0,4.0 | − | >3 months |
| 2/5/2016 | 08:45−08:55 | TMRX−ET / TMRX−ET-1 | 4.5, 5.5, 7.5 | − | >3 months |
| 3/5/2016 | 08:45−08:55 | WM−KLY / WM−KLY-2 | 1, 2, 4 | − | >3 months |
| 25/5/2016 | 08:30−08:40 | BARE−MSD / BARE−MSD-1 | 1.25, 2.5, 5 | − | 1−2 |
| 26/5/2016 | 08:30−08:40 | BARE−MSD / BARE−MSD-2 | 1.25, 2.5, 5 | − | 2−3 |
| 30/5/2016 | 12:00−12:10 | WM−ET / TMRX−ET-2 | 4.5, 5.5, 7.5 | − | >3 months |
| 31/5/2016 | 12:00−12:10 | BARE−MSMR / BARE−MSMR-2 | 2.5, 4.5, 7 | − | >3 months |
| 11/7/2016 | 12:00−12:20 | BARE−MSD / BARE−MSD-3 | 1.25, 2.5, 5 | − | >3 months |
| 11/7/2016 | 18:00−18:20 | BARE−MSD / BARE−MSD-4 | 1.25, 2.5, 5 | − | >3 months |
| 21/2/2017 | 11:20−11:40 | COAST−TKM-SD / COAST−TKM-SD-w | 1, 2.5, 6.5 | +/− | 5 |
| 22/2/2017 | 11:00−11:20 | COAST−TKM-LD / COAST−TKM-LD-w | 1.5, 3, 7 | − | 6 |
| 28/2/2017 | 11:20−11:40 | COAST−EGD-SD / COAST−EGD-SD-w | 1, 2.5, 6.5 | + | 0 |
| 1/3/2017 | 11:07-11:27 | COAST−EGD-MD / COAST−EGD-MD-w | 1, 2.5, 6.5 | +/− | >3 months |
| 2/3/2017 | 11:00-11:20 | COAST−EGD-LD / COAST−EGD-LD-w | 1, 2.5, 6.5 | − | >3 months |
| 2/3/2017 | 12:55-13:15 | SEA−KDM / SEA−KDM-w | 1 | + | >3 months |
| 25/4/2017 | 11:30-11:50 | COAS-EGD-SD / COAST−EGD-SD-s | 1, 2.5, 6.5 | + | >3 months |
| 26/4/2017 | 11:00-11:20 | COAST−EGD-MD / COAST− EGD-MD-s | 1, 2.5, 6.5 | +/− | >3 months |
| 27/4/2017 | 11:00-11:20 | COAST−EGD-LD / COAST−EGD-LD-s | 1, 2.5, 6.5 | − | >3 months |
| 3/5/2017 | 12:10-12:30 | COAST−TKM-SD / COAST−TKM-SD-s | 1, 2.5, 6.5 | − | >3 months |
| 4/5/2017 | 10:30-10:50 | COAST−TKM-LD / COAST−TKM-LD-s | 1.5, 3, 7 | − | >3 months |
| 4/5/2017 | 12:30-12:50 | SEA−KDM / SEA−KDM-s | 1 | + | >3 months |


[a] The suffixes "s" and "w" refer to samplings during spring and winter, respectively. "SD", MD", and "LD" refer to
relatively short, medium, and long distance from the coastline, respectively (see Sect. 2.1). [b] "+", "−" and "+/−"
respectively indicate that the samplings could be, could not be, or may be influenced by emission from the



seawater. [c] Values indicate the number of days before sampling during which precipitation occurred. Additional
abbreviations: MSD, Masada; MSMSR, Mishmar; KLY, Kalya; ET, Ein-Tamar; KDM, Kedem; EGD, Ein-Gedi;
BARE, bare soil site; COAST, coastal soil-salt mixture site; WM, cultivated watermelon site; TMRX, natural
Tamarix site; SEA, sampling near the seawater (see Sect. 2.1.1).

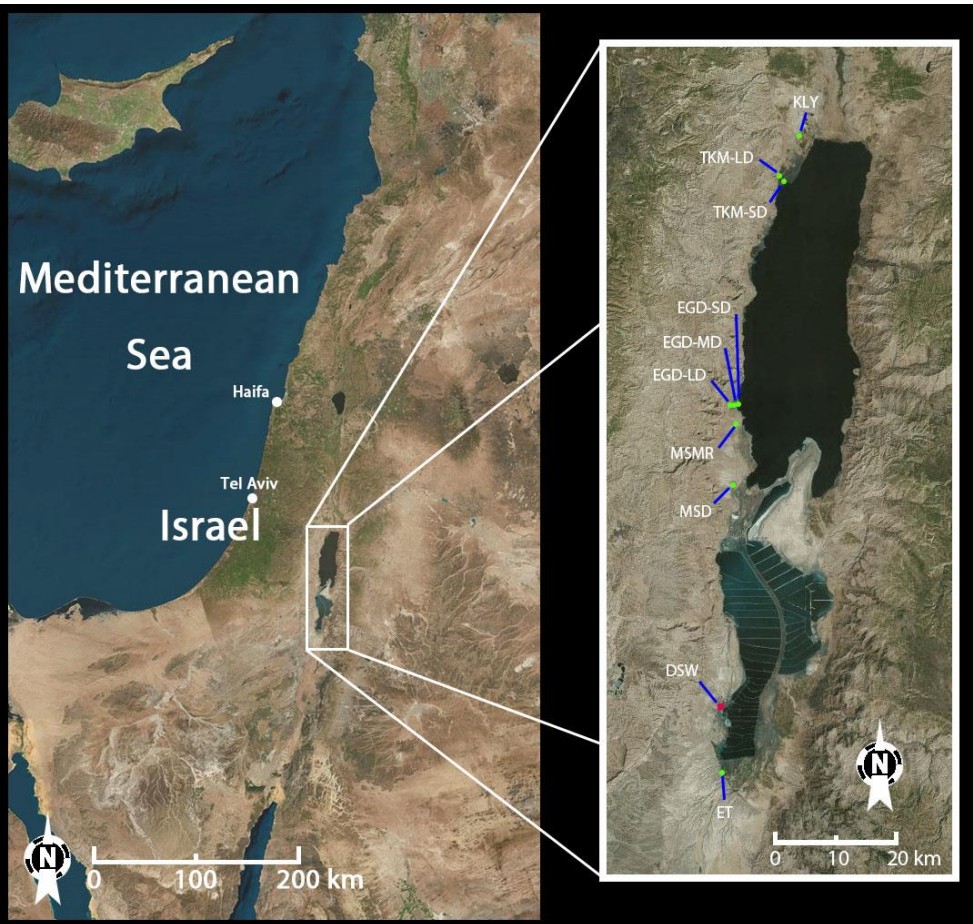

**Fig. 1**. Location and satellite image of the Dead Sea measurement sites and Dead Sea Works (DSW). Left: location
of the Dead Sea. Right: zoom-in of the area of measurement sites.

**2.2 Vertical flux evaluation**
The vertical flux, $F_c$, of a species c, was evaluated according to the gradient approach using the
vertical gradient of c, $\frac{\partial c}{\partial z}$, and a constant, $K_c$:





$$F_c \equiv -K_c \frac{\partial C}{\partial z} \qquad (1)$$

$K_c$ represents the rate of turbulent exchange in Eq. 1 and was evaluated on the basis of the
Monin−Obukhov similarity theory (MOST) described by Lenschow (1995):
$$K_{C_{(z)}} = u_* K Z \phi_C(\zeta) \qquad (2)$$

where $u_*$ is the friction velocity, K is the Von Kármán constant, Z is the measurement height
and $\phi_c$ is a universal function of the dimensionless parameter $\zeta$. According to MOST, vertical
fluxes in the surface layer can be evaluated on the basis of the dimensionless length parameter,
$\zeta$, according to
$$\zeta = (z - d)/L \qquad (3)$$

where z, d and L are the vertical coordinate, zero displacement, and the Monin−Obukhov length,
respectively (Schmugge and André, 1991).
We relied on the commonly used assumption that $\phi_C$ is similar to $\phi_h$ for chemical species
with a relatively long lifetime (Dearellano et al., 1995), and calculated $\phi_h$ using the following
equation for the relationship between $\phi_h$ and $\zeta$, which was found to be valid for $0.004 \leq - z/L \leq 4$
(Dyer and Bradley, 1982; Yang et al., 2001):
$$\phi_h = (1 - 14\zeta)^{-1/2} \qquad (4)$$

We derived L from the Pasquill and Gifford stability class (Pasquill and Smith, 1971) and
roughness length ($z_0$) according to Golder D. (1972). $z_0$ was evaluated based on the specific
surface characteristics at each site using information provided by the WMO (2008). The stability
class was evaluated using the in-situ measured solar radiation and wind speed (Gifford,
2000; Pasquill and Smith, 1971). $u_*$ was derived from the logarithmic wind profile according to
MOST, using the following equation:





$$u(z) = \frac{u_*}{k} \ln(\frac{z-d}{z_0}) \qquad (5)$$


where u(z) is the wind speed at height z, and $\psi_m$ is a correction for diabatic effect on momentum
transport. Using the measured u at a height of 10 m, we calculated the wind speed at each
measurement height according to Gualtieri and Secci (2011):

$$u_2 = u_1 \frac{\ln(z_2/z_0) - \psi m(z_2/L)}{\ln(z_1/z_0) - \psi m(z_1/L)} \qquad (6)$$


where $\psi_m$ is calculated using:

$$\Psi m(Z/L) = 2\ln(1 + X/2) + \ln(1 + X^2)/2) - 2\arctan(X) + \pi/2 \qquad (7)$$


and $\qquad X = (1 - 15\left(\frac{Z}{L}\right))^{1/4} \qquad (8)$

**279 2.3 Soil analyses**

Soil samples at each site were collected up to a depth of 5 cm during summer, at least 3 months
following any rain event in the Dead Sea area. The samples were analyzed for bromine,
chlorine, iodine, organic matter, moisture and Fe in the soil, as well as for pH of the soil. Prior
to halide quantification, extractions for each sample were prepared using $HNO_3$. Total Br and I
were quantified using inductively coupled plasma mass spectrometry (ICPMS). Total Cl was
quantified by potentiometric titration against $AgNO_3$.
To quantify Fe in the soil, microwave-assisted digestion with reverse aqua regia was used,
and Fe concentration was determined by inductively coupled plasma optical emission
spectrometry (ICP−OES). A batch of each sample (~300 mg of dry soil) was digested in reverse
aqua regia ($HNO_3$ (65 %) : HCl (30 %); 3:1 mixture). Digestion was allowed to proceed in
quartz vessels using a "Discover" sample digestion system at high temperature and pressure
(CEM Corporation, NC, USA). The vessels were cooled and the volume was made up to 20 mL
with deionized water. Element concentrations were measured in clear solutions using High
Resolution dual-view ICP−OES PlasmaQuant PQ 9000 Elite (Analytik Jena, Germany). The



reported values represent a low−limit, because the samples were not completely dissolved. Soil
water content and organic matter (OM) were determined by weight loss under dry combustion at
105 °C and 400 °C, respectively. Soil pH was measured in 1:1 soil-to-water extracts with a
model 420 pH-meter (Thermo Orion, Waltham, MA, USA).


**3  Results and discussion**
**3.1  VHOC flux and mixing ratio**
We compared the measured mixing ratios and fluxes with corresponding available information.
Overall, measurements at the Dead Sea boundary layer revealed that the mixing ratios for all
investigated VHOCs were higher than their background MBL levels, pointing to significant
local emissions. No association was observed between the measured mixing ratios and the air
masses flowing from the direction of the Dead Sea Works, which is located to the north-west of
the TMRX−ET site and to the south of all other measurement sites (see Fig. 1), and is the main
anthropogenic source in the area under investigation. The absence of any such association points
to the dominance of natural sources for the VHOCs in the studied area. Table 2 presents a
comparison between measured mixing ratios at the different measurement sites and reported
values for the global MBL. The values indicate that median mixing ratios at the Dead Sea are
higher than corresponding mixing ratios in the MBL by factors of 1.2−8.0 for brominated and
chlorinated VSLS and ~1.5, 1.3 and 1.1 for $CH_3I$, $CH_3Br$ and $CH_3Cl$, respectively. Moreover, as
described below, measured mixing ratios at the Dead Sea were generally also higher than in
coastal areas.
Owing to their large contribution to stratospheric bromine, $CHBr_3$ and $CH_2Br_2$ are the most
extensively studied VSLS in the MBL (Hossaini et al., 2010). The mixing ratios of $CHBr_3$ and
$CH_2Br_2$ that we measured at the Dead Sea ranged from 1.9 to 22.6 pptv and from 0.7 to 18.6
pptv, respectively, which are higher than most of their reported mixing ratios in coastal areas



where the highest mixing ratios have typically been measured. For example, Carpenter et al.
(2009) reported elevated mixing ratios for $CHBr_3$ and $CH_2Br_2$ along the eastern Atlantic coast
ranging from 1.9 to 4.9 and from 0.9 to 1.4 ppt, respectively, and Nadzir et al. (2014) reported
mixing ratios of 0.82−5.25 pptv and 0.90−1.92 ppt for $CHBr_3$ and $CH_2Br_2$, respectively, for
several tropical coastal areas including the Strait of Malacca, the South China Sea and Sulu-
Sulawesi Seas. Somewhat higher mixing ratios for $CHBr_3$ have been measured in only a few
locations, including some at coastal areas near New Hampshire (Zhou et al., 2008), San
Cristobal Island (Yokouchi et al., 2005;O'Brien et al., 2009), Cape Verde (O'Brien et al., 2009),
Borneo (Pyle et al., 2011) and Cape Point (Kuyper et al., 2018;Butler et al., 2007), where the
range (and average) concentrations at those locations were 0.2−37.9 pptv (5.6−6.3), 4.2−43.6
pptv (14.2), 2.0−43.7 pptv (4.3−13.5), 2−60pptv (−) and 4.4−64.6 pptv (24.8), respectively. For
$CH_2Br_2$, the corresponding mixing ratios were reported as 1.3–2.3 pptv, 0.5−4.1 pptv and
0.7−8.8 pptv in New Hampshire, San Cristobal Island and Cape Verde, respectively, which are
comparable with the mixing ratios measured at the Dead Sea.
Figure 2 presents the measured fluxes of all VHOCs studied. On average, the net fluxes of
all measured species, except $C_2HCl_3$ and $CH_3I$, were positive at most of the investigated sites.
The flux magnitudes for $CHBr_3$ and $CH_2Br_2$ were higher than for most reported emissions at the
MBL, but in most cases were smaller than the corresponding average fluxes estimated by Butler
et al. (2007) for global coastal areas (~220 and 110 nmol $m^{-2}$ $d^{-1}$), respectively. In some cases,
however, the fluxes of both species were higher than these values.
$CHCl_3$ emission rates were positive for most measurements and particularly high for
TMRX−ET-2 (213 nmol $m^{-2}$ $d^{-1}$), COAST−EGD-SD-s (883 nmol $m^{-2}$ $d^{-1}$), and
BARE−MSMR-1 (247 nmol $m^{-2}$ $d^{-1}$) (see Yi et al., 2018). For comparison, the emission from
BARE−MSMR-1 is similar to the maximum emission found for tundra peat by Rhew et al.
(2008), while the averaged emissions from COAST−EGD-SD-s and TMRX−ET-2 are higher
than those from temperate peatlands (~496 nmol $m^{-2}$ $d^{-1}$ as measured by Dimmer et al. (2001)).



Whereas emissions during COAST−EGD-SD-s and TMRX−ET-2 might have been affected by
vegetation and seawater, respectively, the emission from BARE−MSMR can be completely
attributed to soil. The latter emission flux in BARE−MSMR is higher than the maximum
emission rate in arctic and subarctic soils (~115 nmol m$^{-2}$ d$^{-1}$) reported by Albers et al. (2017).
Average calculated fluxes for the additional brominated VSLS, CHBr$_2$Cl, and CHBrCl$_2$
were positive for all sites except for CHBr$_2$Cl at COAST−TKM. The mixing ratios of CHBr$_2$Cl
and CHBrCl$_2$ were higher by factors of ~4−14 and ~5−11, respectively, than the average
reported values for the MBL and were also higher than measured mixing ratios in nearby coastal
areas, except for the extremely high CHBr$_2$Cl mixing ratios emitted from a rock pool at Gran
Canaria (ranging from 19 to 130 ppt; (Ekdahl et al., 1998)). For example, Brinckmann et al.
(2012) found mean mixing ratios for CHBr$_2$Cl and CHBrCl$_2$ in coastal areas at the Sylt Islands
(North Sea) of up to 0.2 and 0.1 ppt, respectively, while Nadzir et al. (2014) found CHBr$_2$Cl and
CHBrCl$_2$ mixing ratios of 0.07–0.15 ppt and 0.15–0.22 ppt, respectively, in the tropics. The
measured fluxes that we obtained for CHBr$_2$Cl at the Dead Sea are also higher than the reported
values of 0.8 (range, -1.2−10.8) nmol m$^{-2}$d$^{-1}$ at coastal areas sampled during the Gulf of
Mexico and East Coast Carbon cruise (GOMECC), (Liu et al., 2011). Typically, the CHBrCl$_2$
net flux at the Dead Sea is significantly higher than corresponding fluxes from arctic and
subarctic soils, as recently reported by Albers et al. (2017) (see Fig. 2).
The CH$_3$Cl flux at the Dead Sea was positive for only half of the measurements, while a net
positive flux for all measurements was obtained only at COAST−TKM. The highest positive
fluxes were measured at COAST−EGD and COAST−TKM, with maximum net fluxes of
~10800 and 4900 nmol m$^{-2}$ d$^{-1}$, respectively. These fluxes are comparable in magnitude to those
reported for several terrestrial sources, such as tropical forests (~4520 nmol m$^{-2}$ d$^{-1}$) by
Gebhardt et al. (2008) or by Yokouchi et al. (2002) and for other tropical or subtropical
vegetation (Yokouchi et al., 2007), and they are higher than emissions from dryland ecosystems
including shortgrass steppe or shrublands (Rhew et al., 2001). In some cases the measured



fluxes were higher than average emissions from salt marshes (e.g., ~7300 nmol m$^{-2}$ d$^{-1}$;
(Deventer et al., 2018)), but significantly smaller than the maximum fluxes (e.g., 570000 nmol
m$^{-2}$ d$^{-1}$; (Rhew et al., 2000)).

In contrast to CH$_3$Cl, emissions of CH$_3$Br at the Dead Sea were significantly lower than the

average reported emissions from marshes (e.g., ~600 nmol m$^{-2}$ d$^{-1}$; (Deventer et al., 2018). The
fluxes measured at the Dead Sea were also lower than the reported emission from a coastal
beach in a Japanese archipelago island (~53000 nmol m$^{-2}$ d$^{-1}$), but higher, in most cases, than in
other dryland ecosystems (see Rhew et al. (2001)).

The net flux of CH$_3$I measured at the Dead Sea was negative in 60 % of the measurements.

Positive measured net fluxes of this compound were in most cases comparable to other reported
fluxes over soil and vegetation. For example, Sive et al. (2007) reported a CH$_3$I flux of ~18.7
nmol m$^{-2}$ d$^{-1}$ over soil and vegetation at the AIRMAP Observing Station at Thompson Farm,
NH, USA, and a somewhat lower emission (~12.6 nmol m$^{-2}$ d$^{-1}$) at Duke Forest, NC, USA.
While the elevated flux during COAST−EGD-SD-s (17.0 nmol m$^{-2}$ d$^{-1}$) could potentially have
been affected by flow of the sampled air over the seawater, the positive net fluxes in
BARE−MSMR (1.00 and 4.42 nmol m$^{-2}$ d$^{-1}$) indicate significant emission from bare soil at the
Dead Sea. The emission rates in BARE−MSMR are similar to the measured soil-emission fluxes
of CH$_3$I reported by Sive et al. (2007) at Duke Forest, averaging ~0.27 nmol m$^{-2}$ d$^{-1}$ (range, ~
0.11−4.1 nmol m$^{-2}$ d$^{-1}$).

Most of the sites were found, on average, to be a sink for C$_2$HCl$_3$, which may suggest that

the elevated mixing ratios for this species in the Dead Sea area mostly result from anthropogenic
emission.





**Table 2**. Comparison of VSLS and methyl halide mixing ratios (in pptv) measured at the Dead Sea with their
corresponding values at the marine boundary layer (MBL). Unless otherwise specified, the table presents median,
minimum and maximum VHOC mixing ratios measured at different sites at the Dead Sea (see Table 1 for site
abbreviations) and in the MBL, as reported by Carpenter and Reimann et al. (2014).

| Species | Median/Range | BARE−MSMR | BARE−MSD | COAST−EGD | COAST−TKM | TMRX−ET | WM−KLY | SEA−KDM | All Sites | EF. MBL[a] |
|---|---|---|---|---|---|---|---|---|---|---|
| $CHBr_3$ | Median | 11.3 | 11.0 | 8.0 | 2.6 | 4.7 | 3.1 | 11.0 | **6.2** | **5.2** |
| | Range | 5.6−16.3 | 6.0−22.6 | 4.4−16.8 | 3.6−1.9 | 2.9−7.1 | 2.3−3. 5 | 5.4−16.5 | **1.9−22.6** | **4.8–5.7** |
| $CH_2Br_2$ | Median | 0.9 | 2.7 | 1.8 | 0.8 | 1.3 | 1.1 | 1.4 | **1.1** | **1.22** |
| | Range | 0.9−1.0 | 0.7−18.6 | 0.9−5.1 | 0.9−0.8 | 0.9−1.6 | 1.0−1.2 | 1.1−1.7 | **0.7−18.6** | **1.2–11** |
| $CHBr_2Cl$ | Median | 4.8 | 4.1 | 2.2 | 1.2 | 2.2 | 0.9 | 1.3 | **2.4** | **8** |
| | Range | 3.2 −5.4 | 2.2−11.0 | 0.4−6.5 | 7.5−0.5 | 0.6−3.6 | 0.5−1.0 | 0.5−2.2 | **0.4−11.0** | **4–14** |
| $CHBrCl_2$ | Median | 2.6 | 2.5 | 1.6 | 1.0 | 2.4 | 0.9 | 2.0 | **1.4** | **4.7** |
| | Range | 2.6−3.7 | 0.9−9.6 | 1.0−3.0 | 0.5−1.4 | 0.7−3.9 | 0.6−1.1 | 1.3−2.7 | **0.5−9.6** | **5−11** |
| $C_2HCl_3$ | Median | 1.15 | 1.7 | 1.3 | 1.6 | 1.1 | 2.7 | 1.2 | **1.5** | **3** |
| | Range | 1.22-0.84 | 1.0−2.7 | 0.3−10.5 | 0.4−2.9 | 0.4−1.5 | 1.0−4.1 | 0.8−1.6 | **0.4−10.5** | **8−5.3** |
| $CHCl_3$ | Median | 16.9 | 19.8 | 18.2 | 18.7 | 19.0 | 19.8 | 17.3 | **18.63** | **2.5** |
| | Range | 15.9−20. 5 | 18.8−25.3 | 14.5−27.9 | 15.4−20.1 | 18.4−57.2 | 18.8−5.3 | 16.5−18.2 | **14.5−57.2** | **2.0−7.3** |
| $CH_3I$ | Median | 0.8 | 1.3 | 1.2 | 1.5 | 1.3 | 1.1 | 1.4 | **1.2** | **1.5** |
| | Range | 0.8−0.8 | 1.0−1.5 | 0.4−2.1 | 1.5−1.2 | 0.8−2.8 | 0.7−1.6 | 1.2−1.6 | **0.4−2.8** | **1.3− 1.3** |
| $CH_3Br$ | Median | 8.72 | 10.3 | 8.7 | 8.4 | 10.0 | 8.4 | 9.7 | **9.1** | **1.3 [b]** |
| | Range | 8.1−9.4 | 7.8−13.8 | 7.5−13.3 | 8.5−7.5 | 8.3−13.8 | 7.8−9.2 | 9.1−10.2 | **7.5−13.8** | **N/A** |
| $CH_3Cl$ | Median | 596 | 571 | 595 | 580 | 623* | 643 | 596 | **601** | **1.1 b** |
| | Range | 583−608 | 549−672 | 531−732 | 583−545 | 581−685* | 591−668 | 583−608 | **531−732*** | **N/A** |


[a] Values represent the enrichment factor (EF) at the Dead Sea, reflected as the ratio between the median measured
mixing ratios and the corresponding median values for the MBL; [b]EF is calculated using the annual average for
2012 based on flask measurements by the US National Oceanic and Atmospheric Administration (NOAA)
(http://www.esrl.noaa.gov/gmd/dv/site/ and in-situ measurements by the Advanced Global Atmospheric Gases
Experiment (AGAGE (http://agage.eas.gatech.edu/). See Table 1 for site abbreviations. *Calculation excludes one
$CH_3Cl$ measurement in TMRX−ET-1 (see Sect. 2.1.2).

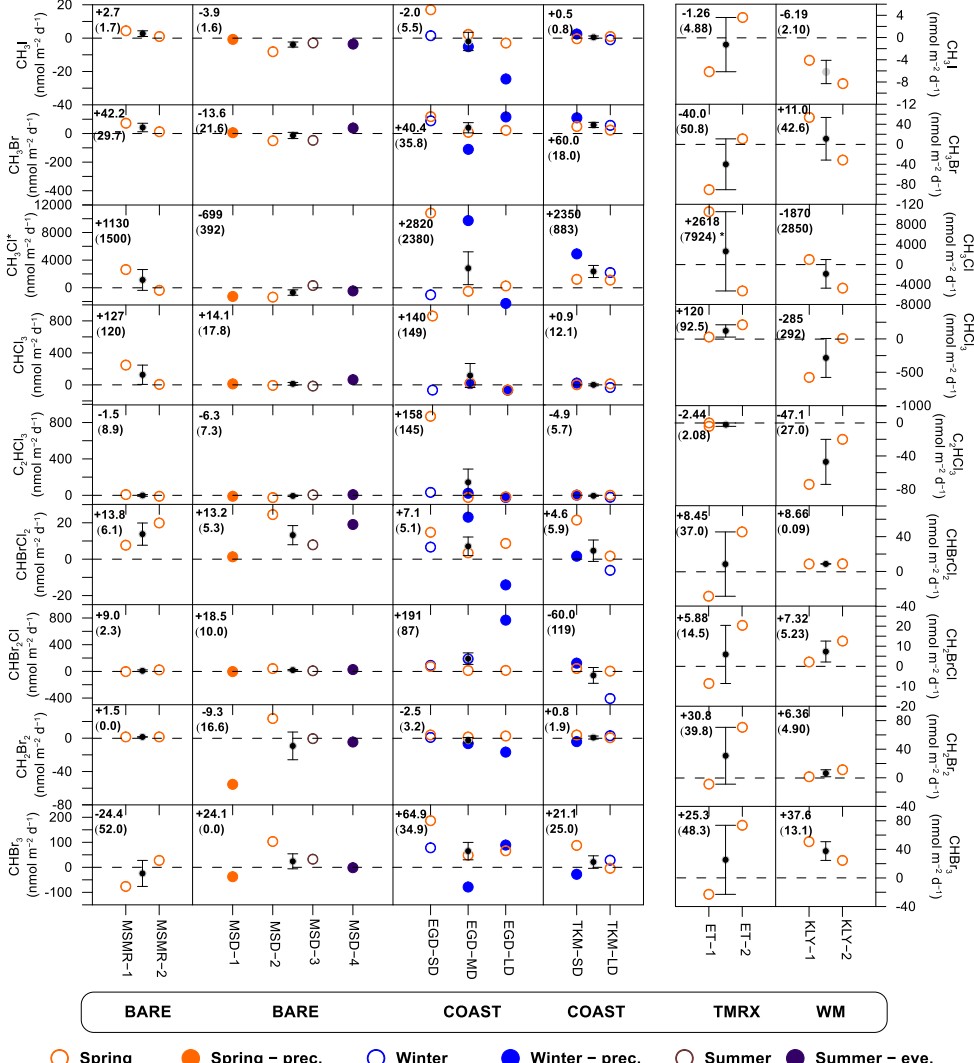

**Fig. 2. VHOCs fluxes at the different measurement sites**. Fluxes are marked by circles to individually indicate measurements during spring, winter, summer, up to 3 days after a rain event in spring ("Spring-prec."), up to 6 days after a rain event in winter ("Winter-prec.") and in summer during evening ("Summer-eve"); for more information about measurement conditions see Table 1. Black filled circles and error bars respectively represent the average and standard error of the mean (SEM) for each measurement site. Dashed lines represent zero flux. Mean flux value and standard deviation (SD; in parenthesis) are shown for each site and species. See Table 1 for measurement sites and measurement abbreviations. * Calculation of mean flux and SD excludes one $CH_3Cl$ measurement in TMRX−ET-1 (see Sect. 2.1.2).



### 3.2 Factors controlling the flux of VHOCs

### 3.2.1 Seasonal, meteorological and spatial effects

The results presented in Sect. 3.1 record elevated mixing ratios and net fluxes for all investigated VHOCs, with relatively less frequent positive fluxes for $CH_3I$ and $C_2HCl_3$. All of the investigated VHOCs except $C_2HCl_3$ were associated with a positive average net flux from at least one of the two bare soil sites BARE−MSMR and BARE−MSD (Fig. 2), and for all VOHCs, except $C_2HCl_3$ and $CH_3Cl$, all measured mixing ratios were highest over at least one of these bare soil sites (Table 2). These findings suggest that a significant emission for all of the investigated VHOCs occurred from bare soil located within at least a few kilometers from the Dead Sea water.

No clear impact of meteorological conditions on the measured net flux rates or mixing ratios was observed. We could not identify any clear association between flux magnitude and any parameter, including solar radiation intensity, measurement time, temperature, or daytime relative humidity.

Our findings on the effects of season and distance from the sea on the measured fluxes are presented in Fig. 2, which presents the results of our measured fluxes for spring and winter and for different distances from the sea at COAST−EGD and COAST−TKM. It can be seen from the figure that whereas for most compounds there were no clear differences in fluxes between spring and winter, the measured fluxes for $CH_3I$, $CHBrCl_2$ and $CH_2Br_2$ were generally higher in the spring. No clear impact of the distance from the seawater on the measured net fluxes could be detected, including in cases where a significant fraction of the footprint included the seawater, such as during COAST−TKM-SD-w and, to a lesser extent, during COAST−EGD-SD-w and COAST−EGD-SD-s. However, as discussed below (see Sects. 3.2.2, 3.2.3), owing to variations in soil properties the emissions near the seawater tended to be more frequent and more intense. Figure 3 compares the mixing ratios of the measured VHOCs at different distances from the seawater, and individually for winter and spring. No clear impact of season



or distance from the seawater on the mixing ratios can be discerned in this figure, also for the
sampling over SEA−KDM which directly represented air masses over the seawater (Sect. 2.1.1).
Nevertheless, further investigation, using direct flux measurements over the Dead Sea water, is
needed to study the potential emission of VHOCs from the Dead Sea water.

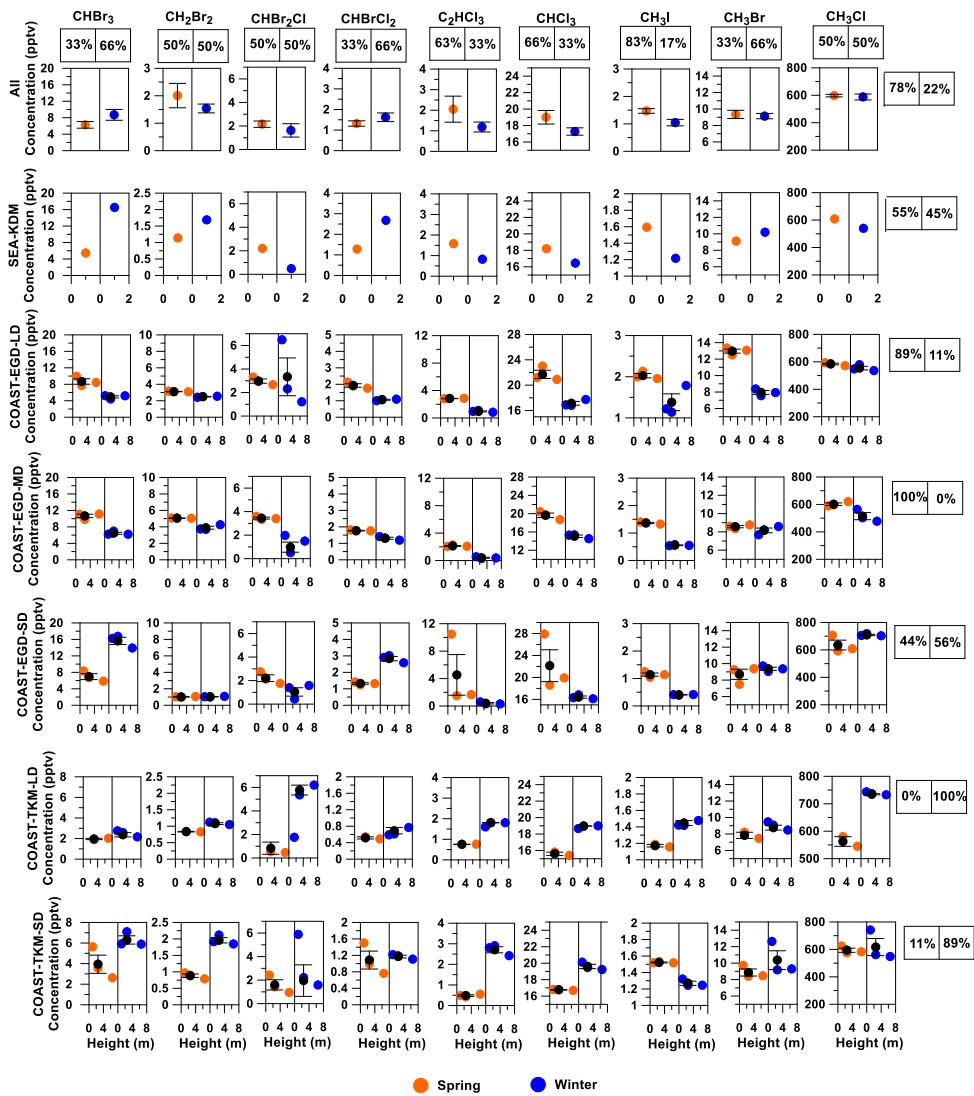

**Figure 3. Seasonal and spatial influences on measured mixing ratios of VHOCs.** Measured VHOC mixing
ratios are presented vs. vertical height above surface level, separately for winter (blue) and spring (orange). Black
filled circles and error bars represent the average and SEM, respectively. Values above and to the right of the
figure indicate the percentage of time during which average mixing ratios were higher during spring (left box) or





during winter (right box), individually for TKM, EGD and SEA−KDM sites, and for all of these sites together
(All), for all sites and all species, respectively (see Table 1 for measurement site abbreviations).

**3.2.2 Impact of specific site characteristics and ambient conditions**
The formation of VHOCs requires a chemical interaction between organic matter and halides,
induced by biogeochemical, biochemical, or macrobiotic processes (Kotte et al., 2012;Breider
and Albers, 2015). Despite the extreme salinity, biotic activity was detected both in the water
and in the soil of the Dead Sea (see Sect. 1), demonstrating that biotic activity can potentially
contribute to VHOC emission in this area. Previous studies on emission of VHOCs from soil
and sediments revealed that organic matter content and type, halide ion concentrations, pH, and
the presence of an oxidizing agent (most frequently referred to Fe (III)) also play important roles
in the emission rate of VHOCs (see (Kotte et al., 2012)).
Table 3 provides a basic representation of these parameters. The table records substantial
enrichment of Cl and Br in the sites closest to the seawater (COAST−EGD-SD and
COAST−TKM-SD) and lower concentrations at larger distance from the seawater. For
comparison, both Br and Cl concentrations are much higher than those reported by Kotte et al.
(2012) for various saline soils and sediments (0.12−0.32 g kg$^{-1}$ and 6.1−120 g kg$^{-1}$,
respectively), but are lower in BARE−MSMR and BARE−MSD for Br and for both Cl and Br
in WM−KLY. No enrichment of I in the soil samples was observed (e.g., see Keppler et al.
(2000);Kotte et al. (2012)). The OM content of the samples is generally higher than would be
expected in desert soil. For comparison, forest floors typically contain 1−5 % OM (Osman,
2013). Detection of VHOC emissions from the soil was in some cases associated with higher
soil OM (e.g., Albers et al., 2017; Kepller et al., 2000) and in some cases with lower soil OM
(e.g., Kotte et al., 2012; Hubber et al., 2009) than reported here. Table 3 provides only an
underestimated value of Fe, rather than Fe (III), in the samples. Note, however, that similar soil
Fe content as reported here as a low-limit value corresponded with the finding of small amounts
of VHOCs emission, while the emission rates became saturated when enrichment with Fe (III)



was relatively minor (Keppler et al., 2000). Saturation at relatively low soil Fe concentrations was also reported by Huber et al. (2009). Hence, variations in Fe across different sites may result in different emission rates.

Table 4 merges the mixing ratios, fluxes, and the ratio between flux and corresponding mixing ratio (F:C) during all measurement periods, for all investigated VHOCs. F:C is used to study the potential contribution of each site to the VHOC mixing ratios measured at that site. The number of samplings at each site was limited, but Table 4 indicates that the fluxes measured at some of the sites were relatively high. In both COAST−TKM and COAST−EGD sites we observed relatively high frequencies of elevated fluxes, particularly from the SD sites, and to some extent also in COAST−EGD-MD. Moreover, for both COAST−EGD and COAST−TKM, during both spring and winter the occurrence of positive fluxes was correlated with proximity to seawater (i.e., COAST−EGD-SD > COAST−EGD-MD > COAST−EGD-LD, and COAST−TKM-SD > COAST−TKM-LD). All of these sites contain mixtures of soil and salt-deposited structures (see Sect. 2.1.1), and Table 3 indicates that soil concentrations of both Br and Cl correlated with proximity to seawater at both COAST−EGD and COAST−TKM. The concentration of I in the soil showed a similar trend only in COAST−TKM (see Table 3). This association between the positive net flux magnitude and incidence and the soil halide concentrations points to an increase in VHOC emission with salinity, even under the hypersaline conditions of the Dead Sea area. This interpretation is supported by the fact that whereas for COAST−TKM-SD both soil water and OM content were relatively high, for COAST−EGD-SD no other parameter, except for the halides soil concentration, was clearly higher for COAST−EGD-MD and COAST−EGD-LD. The generally higher emission rates for COAST−EGD than for COAST−TKM (Table 4) may suggest, in view of the apparently lower Fe content for COAST−EGD (Table 3), that the emission of VHOCs from these sites was not significantly limited by the availability of Fe (III) in the soil.



Our measurements revealed no clear contribution of vegetation to the emission fluxes or the
mixing ratios (Fig. 2 and Table 2), but it should be emphasized that our ability to define their
role in VHOC emission and uptake in this study was limited. Table 4 indicates relatively high
positive net fluxes for several species in one out of the two measurements at each of the
vegetated sites TMRX−ET and WM−KLY. Particularly for TMRX−ET-2, emissions were high
for all of the investigated VSLS except $C_2HCl_3$ and $CH_3Cl$ (Fig. 2, Table 4).
Whereas during COAST−EGD-SD-s all measured emission fluxes were positive and high,
emissions during COAST−EGD-SD-w were generally lower and were negative for $CH_3Cl$ and
$CHCl_3$. Based on the wind direction, in both cases the sampling footprint included both the
seawater and a narrow strip of bare soil mixed with salty beds (estimated at about 40 % of the
footprint), very close to the seawater. The main notable difference between the two
measurement days is that precipitation occurred just before COAST−EGD-SD-w, while there
was no precipitation event for several weeks prior to COAST−EGD-SD-s. Note that the much
higher fluxes during COAST−EGD-SD-s than during COAST−EGD-SD-w did not result in a
proportional increase in F:C (e.g., for $C_2HCl_3$), and for some species the F:C for COAST−EGD-
SD-s was even lower than for COAST−EGD-SD-w (e.g., for $CHBr_3$ and $CH_3Br$). This
decoupling between fluxes and mixing ratios may be attributable to the fact that flux and
concentrations can have very different footprints, such that under a widespread rain event the
mixing ratios at COAST−EGD-SD might be less directly affected by the local changes in the net
fluxes.
A less widespread and more spatially limited rain event occurred ~1.5 and ~2.5 days before
BARE−MSD-1 and BARE−MSD-2, respectively. It is notable that the emission fluxes for
BARE−MSD-1 are lower and more negative for most of the species than those for
BARE−MSD-3 or BARE−MSD-4. Also, the occurrence of positive net flux increased according
to the order BARE−MSD-1 < BARE−MSD-2 < BARE−MSD-3 < BARE−MSD-4 (see Table 4).
This suggests that increased soil water content caused by rain events can decrease the emission





rates of certain VHOCs. Furthermore, the local rain event in BARE−MSD may be a major

reason for the generally more frequent and higher net fluxes in BARE−MSMR than in

BARE−MSD, and the fact that unlike in the case of COAST−EGD-SD measurements, the low

flux values for BARE−MSD-1 are accompanied, in most cases, by proportionally low F:C

values (Table 4). Interestingly, ~2.5 days after the rain event the measured fluxes at

BARE−MSD-2 were higher for all brominated VSLS but negative for all other VHOCs, which

may indicate the involvement of microbial activity in the emission processes.

The reduction in net flux rates following rain events did not occur for all species, and there

was no clear consistency in this aspect across the two sites BARE−MSD and COAST−EGD-

SD. Thus, further research on the effects of rain on the various VHOCs and ambient conditions

is required. Nevertheless, the analyses clearly demonstrate that strong emission rates do not

depend on rain occurrence, in agreement with findings by Kotte et al. (2012). The lower

emission fluxes following the rain event may be attributable to the low infiltration rate of

VHOCs through the soil, or by salt dilution and washout, or both.

**Table 3.** Soil properties. OM, organic matter; soil water content (SWC); I, Br, Cl and Fe dry weight fraction and
soil pH. See Table 1 for measurement site abbreviations.

| Site | pH | OM (%) | SWC (%) | I mg kg soil dw$^{-1}$ | Br gr kg soil dw$^{-1}$ | Cl gr kg soil dw$^{-1}$ | Fe mg kg$^{-1}$ |
|---|---|---|---|---|---|---|---|
| **BARE−MSMR** | 7.46 | 1.96 | 1.90 | 2.24 | 0.007 | 6.70 | >20800 |
| **BARE−MSD** | 7.41 | 3.61 | 3.61 | 2.79 | 0.027 | 41.2 | >7450 |
| **COAST−EGD-SD** | 7.61 | 2.28 | 1.79 | 0.24 | 1.47 | 202 | >1120 |
| **COAST−EGD-MD** | 7.93 | 0.35 | 0.35 | 0.57 | 0.293 | 37.4 | >3140 |
| **COAST−EGD-LD** | 7.70 | 3.67 | 2.58 | 1.03 | 0.008 | 26.1 | >5950 |
| **COAST−TKM-SD** | 7.43 | 24.1 | 33.7 | 3.19 | 3.93 | 169 | >12500 |
| **COAST−TKM-LD** | 7.80 | 3.40 | 1.64 | 1.14 | 0.186 | 19.5 | >10600 |
| **TMRX−ET** | 7.88 | 3.14 | 2.97 | 2.69 | 0.474 | 85.2 | >10100 |
| **WM−KLY** | 7.64 | 4.10 | 1.40 | 1.69 | 0.013 | 1.12 | >7680 |





**Table 4.** VHOC flux and its correspondence with mixing ratios. Shown are the measured flux (nmol m$^{-2}$ d$^{-1}$, upper
cells; bold) and the ratio between flux and mixing ratio (ppt; F:C; lower cells; italic), obtained for the different
measurements. Also shown are the average flux ("Mean") and average positive flux ("Mean positive") for all
species, as well as the incidence of positive flux (*X*) individually for each site and each VHOC. (See Table 1 for
abbreviations of the different measurements).

| Species Site | CH$_2$Br$_2$ | CHBr$_3$ | CHBr$_2$Cl | CHBrCl$_2$ | CHCl$_3$ | C$_2$HCl$_3$ | CH$_3$Cl | CH$_3$Br | CH$_3$I | X (%) |
|---|---|---|---|---|---|---|---|---|---|---|
| BARE− MSMR-1 | **1.43** | **-76.5** | **-3.27** | **7.68** | **247** | **7.33** | **2629** | **71.9** | **4.42** | 77.8 |
| | *1.59* | *-4.84* | *-0.62* | *2.94* | *13.3* | *6.36* | *4.03* | *8.22* | *5.77* | |
| BARE− MSMR-2 | **1.51** | **27.6** | **21.3** | **19.9** | **6.51** | **-10.4** | **-378** | **12.6** | **1.00** | 77.8 |
| | *1.66* | *4.18* | *5.65* | *6.30* | *0.40* | *-9.83* | *-0.69* | *1.47* | *1.29* | |
| BARE− MSD-1 | **-55.4** | **-37.7** | **-3.58** | **1.32** | **12.1** | **-11.0** | **-1266** | **5.26** | **-0.73** | 33.3 |
| | *-3.30* | *-4.80* | *-1.61* | *1.38* | *0.61* | *-9.00* | *-2.10* | *0.38* | *-0.55* | |
| BARE− MSD-2 | **23.5** | **103** | **41.8** | **24.5** | **-6.02** | **-24.8** | **-1368** | **-50.3** | **-8.14** | 44.4 |
| | *4.89* | *4.95* | *4.01* | *2.66* | *-0.21* | *-10.38* | *-2.14* | *-4.59* | *-7.69* | |
| BARE− MSD-3 | **-0.60** | **32** | **8.69** | **7.92** | **-14.6** | **4.32** | **311** | **-47.9** | **-2.95** | 55.6 |
| | *-0.82* | *5.46* | *2.62* | *3.42* | *-0.96* | *2.38* | *0.56* | *-5.34* | *-2.27* | |
| BARE− MSD-4 | **-4.61** | **-1.41** | **26.96** | **19.1** | **64.7** | **6.39** | **-472** | **38.44** | **-3.58** | 55.6 |
| | *-5.26* | *-0.10* | *5.74* | *7.00* | *3.98* | *3.85* | *-0.85* | *4.37* | *-2.38* | |
| COAST− EGD-SD-w | **0.85** | **78.1** | **90.0** | **6.63** | **-42.8** | **47.3** | **-1040** | **88.4** | **1.45** | 77.8 |
| | *0.05* | *75* | *78.9* | *5.02* | *-2.61* | *118* | *-1001* | *85.2* | *1.40* | |
| COAST− EGD-MD-w | **-6.53** | **-79.0** | **187** | **23.1** | **38.5** | **37.5** | **9719** | **-111** | **-5.16** | 55.6 |
| | *-6.70* | *-12.25* | *141* | *17.63* | *3.08* | *92.0* | *17.3* | *-13.6* | *-9.32* | |
| COAST− EGD-LD-w | **-16.7** | **88.7** | **768** | **-14.2** | **-43.7** | **-8.97** | **-2281** | **116** | **-24.5** | 33.3 |
| | *-6.72* | *17.98* | *230* | *-13.40* | *-2.55* | *-9.41* | *-3.83* | *14.6* | *-17.8* | |
| COAST− EGD-SD-S | **3.71** | **187** | **72.3** | **14.8** | **883** | **884** | **10817** | **118** | **17.0** | 100 |
| | *3.63* | *26.80* | *32.9* | *11.21* | *39.9* | *195* | *17.0* | *13.6* | *14.9* | |
| COAST− EGD-MD-s | **1.35** | **48.6** | **13.4** | **3.42** | **46.4** | **-8.39** | **-530** | **8.10** | **2.27** | 77.8 |
| | *0.27* | *4.54* | *3.88* | *1.93* | *2.36* | *-3.95* | *-0.88* | *0.95* | *1.66* | |
| COAST− EGD-LD-s | **2.52** | **66.0** | **13.8** | **8.68** | **-40.8** | **-2.03** | **261** | **22.3** | **-2.96** | 66.7 |
| | *0.81* | *7.60* | *4.65* | *4.50* | *-1.90* | *-0.72* | *0.45* | *1.72* | *-1.46* | |
| COAST− TKM-SD-w | **-4.15** | **-28.1** | **123** | **1.62** | **22.8** | **0.89** | **4895** | **110** | **2.42** | 77.8 |
| | *-2.12* | *-4.46* | *38.1* | *1.38* | *1.16* | *0.33* | *7.56* | *10.6* | *1.91* | |
| COAST− TKM-LD-w | **2.95** | **28.5** | **-408** | **-6.2** | **-32.9** | **-22.0** | **2200** | **57.3** | **-1.03** | 44.4 |
| | *2.70* | *11.4* | *-91.7* | *-9.33* | *-1.75* | *-12.67* | *3.47* | *6.37* | *-0.72* | |
| COAST− TKM-SD-s | **3.80** | **87.7** | **42.7** | **21.4** | **0.99** | **2.00** | **1210** | **49.3** | **-0.38** | 88.9 |
| | *4.31* | *22.26* | *26.7* | *19.63* | *0.06* | *4.14* | *2.04* | *5.57* | *-0.25* | |
| COAST− TKM-LD-s | **0.56** | **-3.83** | **2.07** | **1.67** | **12.6** | **-0.31** | **1100** | **23.6** | **0.97** | 77.8 |
| | *0.68* | *-1.95* | *4.14* | *3.21* | *0.81* | *-0.42* | *1.96* | *3.01* | *0.83* | |
| TMRX− ET-1 | **-8.93** | **-23.0** | **-8.64** | **-28.5** | **27.6** | **-0.36** | **10500*** | **-90.8** | **-6.14** | 11.1 |
| | *--6.27* | *-7.06* | *-11.5* | *-0.30* | *0.87* | *-0.57* | *12.0* | *-8.46* | *-2.74* | |
| TMRX− ET-2 | **70.6** | **73.7** | **20.4** | **45.4** | **213** | **-4.53** | **-5300** | **10.9** | **3.61** | 77.8 |
| | *62.5* | *11.4* | *5.86* | *12.9* | *11.1* | *-3.23* | *-8.75* | *1.11* | *4.17* | |
| WM− KLY-1 | **1.45** | **50.7** | **2.09** | **8.57** | **-577** | **-74.1** | **983** | **53.5** | **-4.01** | 55.6 |
| | *1.36* | *15.3* | *2.03* | *7.65* | *-27.4* | *-53.1* | *1.53* | *6.30* | *-5.79* | |
| WM− KLY-2 | **11.3** | **24.5** | **12.6** | **8.76** | **6.31** | **-20.0** | **-4730** | **-31.6** | **-8.29** | 66.7 |
| | *10.9* | *8.84* | *19.7* | *13.27* | *0.31* | *-5.44* | *-7.62* | *-3.77* | *-1.36* | |
| | | | | | | | | | | |
| Mean | **1.43** | **32.3** | **51.1** | **8.78** | **41.2** | **40.1** | **1360** | **22.7** | **-1.74** | |
| | **3.84** | **9.01** | **25.0** | **4.95** | **2.03** | **15.2** | **-48.0** | **6.39** | **-1.02** | |
| Mean positive | **9.66** | **68.9** | **90.4** | **13.2** | **122** | **124** | **4060** | **52.4** | **4.14** | |
| | **7.26** | **16.6** | **37.9** | **7.18** | **6.00** | **52.8** | **6.17** | **10.9** | **4.00** | |
| X (%) | **65** | **65** | **80** | **85** | **65** | **40** | **55** | **75** | **40** | |

* Calculation of flux excludes one CH$_3$Cl measurement in TMRX−ET-1 (see Sect. 2.1.2).



### 3.2.3 Factors controlling the flux of specific VHOCs

Trihalomethanes: Table 4 indicates a higher overall incidence of positive flux measured in brominated than in chlorinated VHOCs, except for $CHCl_3$ for which the net flux is positive for 65 % of the measurements. The net flux of the brominated trihalomethanes also tends to be higher than the more chlorinated trihalomethanes ($CHBr_2Cl > CHBr_3 > CHBrCl_2 > CHCl_3$; see Table 4). When averaging over the positive fluxes only, $CHCl_3$ exhibits the second highest flux of all investigated VHOCs (175 nmol m$^{-2}$ s$^{-1}$), suggesting both high emission and their balance to some extent by sinks for this species.

Natural emission of trihalomethanes from soil has been shown to occur without microbial activity involvement, induced via oxidation of organic matter by an electron acceptor such as Fe(III) (Huber et al., 2009) or via hydrolysis of trihaloacetyl compounds (Albers et al., 2017). The soils studied by Albers et al. (2017) are in general significantly richer in OM than the soil at the Dead Sea, except for COAST−TKM-SD. Hence, the apparently higher emission from the Dead Sea soil may indicate either a different mechanism leading to the release of trihalomethanes from soil or only a weak dependency on availability of OM in the soil. The latter possibility may be supported by the fact that Albers et al. (2017) did not find a correlation between chloroform emission rate and organic chlorine in the soil, and by the association found in the present study between soil halide content and VHOC flux for COAST−EGD and COAST−TKM sites (Sect 3.2.2). While trihalomethane formation via organic matter oxidation was reported to occur more rapidly at low pH, and specifically at pH > ~3.5 (Huber et al., 2009;Ruecker et al., 2014), its formation via hydrolysis of trihaloacetyl is expected to occur faster at the relatively high pH of ≥ 7 (Hoekstra et al., 1998;Albers et al., 2017). Yet, according to Ruecker et al. (2014), in hypersaline sediments the formation of VHOCs via organic matter oxidation involving Fe(III) can occur at pH > 8 for biotic processes. Therefore, given the relatively high pH (~7.4−7.9) at these sites, our present findings of high trihalomethane



emission rates from bare and from vegetated soil sites support the evidence supplied by Albers
et al. (2017) concerning the emission of trihalomethanes from the soil after trihaloacetyl
hydrolysis (Table 3). Albers et al. (2017) demonstrated that their proposed mechanism supports
the emission of $CHCl_3$ and $CHBrCl_2$ from soil, and suggested that additional halomethanes with
higher number of bromine atoms can be expected to be emitted via this mechanism, but at much
lower rates. Hence, the higher net fluxes for $CHBr_2Cl$ and $CHBr_3$ at the Dead Sea could occur
either because of the markedly higher composition of different halides in the Dead Sea soil (see
Table 3) or because another mechanism is also playing a role in the emission. The finding of
Hoekstra et al. (1998) that bromine enrichment mainly enhances the emission of $CHBr_3$ and
$CHBr_2Cl$ rather than that of $CHBrCl_2$ supports the former possibility. While both Cl and Br soil
contents are relatively high for the SD and COAST-EGD-MD, where emission of brominated
trihalomethanes was higher than that of chlorinated trihalomethanes (see Table 4), remarkably
high Br/Cl (1:43) relative to other sites was found in COAST−TKM-SD. No clearly more
elevated positive flux of brominated compared to chlorinated trihalomethanes was observed for
this site, suggesting that the main reason for the relatively elevated brominated trihaloethanes at
the SD sites and COAST−EGD-MD is the high Br content rather than the Br/Cl ratio. The
relatively elevated net flux of brominated trihalomethanes from the soil and vegetated sites
indicates that relatively high rates of emission of these species can also occur from soils that are
much less rich in Br than the SD sites and COAST−EGD-MD sites (see Tables 3, 4). Yet, the
emission rates of $CHBrCl_2$ at the Dead Sea were generally higher than those observed by Albers
et al. (2017), probably reflecting the higher chlorine soil content at the Dead Sea.
Methyl halides: Similarly to $CHCl_3$, the methyl halides $CH_3Cl$, $CH_3Br$, and $CH_3I$ exhibit
relatively large differences between their average overall measured fluxes and the average
positive flux, implying high rates of both emission and deposition. The average positive flux of
$CH_3Cl$ is the highest of all the VHOCs investigated, indicating strong emission and deposition
for this species at the Dead Sea. Several studies have indicated that soil tends to act as a sink for



$CH_3Cl$ (Rhew et al., 2003). The relatively high fluxes of $CH_3Cl$ and $CH_3Br$ at WM−KLY-1 (983
and 53.5 nmol m$^{-2}$ d$^{-1}$, respectively) may point to emission of this species from the local
vegetation, in agreement with previous studies (Sect. 1), and potentially caused by a microbial-
or fungal-induced emission (Moore et al., 2005; Watling and Harper, 1998).
A positive net flux for $CH_3I$ was observed at least once at each of the vegetated soils, bare
soils, or soils mixed with salt deposit mixtures (Table 4), but the fluxes we observed were not
significantly higher than those obtained in previous studies (Sect. 3.1), a finding that might be
attributable to the small concentrations of I in the soil relative to those of the other halides. At
Duke Forest, Sive et al. (2007) observed a soil-emission $CH_3I$ flux of ~0.27 nmol m$^{-2}$ d$^{-1}$ on
average (ranging from ~ 0.11 to 0.31 nmol m$^{-2}$ d$^{-1}$) under precipitation conditions in June, and
higher emission rates (0.8 and 4.1 nmol m$^{-2}$ d$^{-1}$) under warmer and dryer conditions in
September. In agreement with those findings, although generally our analyses did not indicate
clear seasonal effects, we found that in all cases the net $CH_3I$ fluxes were higher in spring than
in winter, except for COAST−TKM-SD (Fig. 2). Also, in 83 % of the measurements the $CH_3I$
mixing ratios were higher in spring than in winter (Fig. 3).
Relatively high fluxes of $CH_3Cl$ and $CH_3Br$, and to a lesser extent of $CH_3I$, were observed at
the COAST−TKM and COAST−EGD sites, particularly from the sites closest to the seawater.
According to Keppler et al. (2000), the presence of Fe(III), OM and halide ions are basically
sufficient to result in emission of methyl halides from both soil and sediments by a natural
abiotic process (Sect.1). The strong emission of methyl halide from the COAST−TKM and
COAST−EGD sites indicates that these species can be emitted at high rates from saline soil that
is not rich in OM. The strongest emissions occurred from COAST−TKM-SD, COAST−EGD-
SD and to some extent from COAST−EGD-MD, pointing to a high sensitivity of methyl halide
emission to soil OM and/or halide content (see Table 3). However, the emission of methyl
halides from COAST−TKM-SD, where soil OM is substantially higher than at all other



investigated sites, is similar to or lower than the emission from COAST−EGD-SD and
COAST−EGD-MD.
In a study of emissions of the three methyl halides from soil by controlled experiments,
Keppler et al. (2000) found a decrease in the efficiency of methyl halide emission according to
$CH_3I > CH_3Br > CH_3Cl$ (10:1.5:1; mole fractions). We estimated the emission efficiencies of the
different methyl halides based on the ratio between their fluxes and the concentrations of halide
in the soil. To maintain consistency with the calculations of Keppler et al. (2000) our calculation
was also based on mole fractions, and took into account only positive fluxes, on the assumption
that they are closer in magnitude to emission. This corresponded with measured soil halide
concentration proportions for Cl:B:I as 38487:445:1, and the evaluated emission efficiency
proportions were 57.7:1.56:1 for $CH_3I$, $CH_3Br$ and $CH_3Cl$, respectively. These calculations
verify an increasing efficiency of methyl halide emission such that $CH_3Cl < CH_3Br < CH_3I$, in
agreement with Keppler et al. (2000). These findings suggest that the methylation and emission
of $CH_3Br$ and $CH_3Cl$ in our study were controlled by abiotic mechanisms similar to those
reported by Keppler et al. (2000). The apparently higher relative efficiency of $CH_3I$ emission
may point to emissions of $CH_3I$ via other mechanisms in the studied area, as discussed in Sect.
3.3. It should be noted, however, that we based our calculations on positive flux and not
emission flux, which might also be a reason for the inconsistency between relative emission
efficiency of $CH_3I$ calculated by Keppler et al. (2000) and by us.
$\underline{C_2HCl_3}$: Only COAST−EGD was found to be, on average, a source for $C_2HCl_3$, mostly
derived from strong emissions from COAST−EGD-SD (see, e.g., Fig. 2). COAST−EGD-SD is a
mixture of salt beds with salty soil and therefore the elevated emissions of $C_2HCl_3$ at this site
appear to support previous evidence for the emission of this gas by halobacteria from salt lakes,
as reported by Weissflog et al. (2005). Additional chlorinated VHOCs, including $CHCl_3$ and
$CH_3Cl$, also demonstrated increased emission from this site, in line with the findings of
Weissflog et al. (2005). Note that the net measured fluxes for most of the VHOCs investigated





during COAST−EGD-SD-w were smaller than those of COAST−EGD-SD-s, as discussed in
Sect. 3.2.2.
$\underline{CH_2Br_2}$: $CH_2Br_2$ showed positive fluxes from all site types, with a positive average net flux
from most sites (see Fig. 2). The highest fluxes were observed over TMRX−ET (TMRX−ET-2)
and over bare soil (BARE−MSD-2). Correlation of $CH_2Br_2$ with trihalomethanes will be
discussed in Sect. 3.3

**3.3 Flux and mixing ratio correlations between VHOCs**
Table 5 presents the correlations between the evaluated mixing ratios of VHOCs at the Dead
Sea, separately for all sites and for the terrestrial sites only. In most cases the correlations
between species over the terrestrial sites were low ($r^2 < 0.1$), but were substantially higher for
the brominated trihalomethanes ($CHBr_3$−$CHBrCl_2$ ($r^2 = 0.62$), $CHBr_2Cl$−$CHBrCl_2$ ($r^2 = 0.75$),
and $CHBr_2Cl$−$CHBr_3$ ($r^2 = 0.72$), supporting a common source mechanism for these species.
Relatively high correlations were also obtained, although to a lesser extent, between methyl
halides, particularly between $CH_3Cl$ and $CH_3Br$ ($r^2 = 0.57$). Correlations were in most cases
either similar or smaller, when we included measurements over the seawater site SEA−KDM,
which may reinforce predominant contribution of VHOCs from terrestrial sources in the area of
the Dead Sea. Table 5 shows relatively high correlations of $CHCl_3$ with all the methyl halides
$CH_3Cl$, $CH_3Br$ and $CH_3I$ (with $r^2$ ranging from 0.19 to 0.28), suggesting common emission
sources and/or sinks for these species.



**Table 5.** Correlations between the mixing ratios of VHOCs. Shown is the coefficient of determination ($r^2$) between
each VHOC pair for the measured mixing ratio, when calculated over all sites excluding SEA−KDM and including
SEA−KDM (in parenthesis).

|  | CHBrCl$_2$ | CHBr$_3$ | CHBr$_2$Cl | CHCl$_3$ | CH$_2$Br$_2$ | C$_2$HCl$_3$ | CH$_3$Cl | CH$_3$Br |
|---|---|---|---|---|---|---|---|---|
| **CH$_3$I** | 0.05 (0.05) | 0.02 (0.02) | 0.02 (0.02) | 0.20 (0.21) | 0.01 (0.01) | 0.03 (0.03) | 0.09 (0.10) | 0.13 (0.13) |
| **CH$_3$Br** | 0.04 (0.03) | 0.14 (0.14) | 0.07 (0.06) | 0.19 (0.18) | 0.27 (0.26) | 0.04 (0.03) | 0.57 (0.56) | |
| **CH$_3$Cl\*** | 0.00 (0.00) | 0.05 (0.04) | 0.02 (0.02) | 0.28 (0.28) | 0.01 (0.01) | 0.01 (0.01) | | |
| **C$_2$HCl$_3$** | 0.01 (0.01) | 0.03 (0.01) | 0.02 (0.03) | 0.07 (0.07) | 0.00 (0.00) | | | |
| **CH$_2$Br$_2$** | 0.00 (0.00) | 0.05 (0.04) | 0.01 (0.01) | 0.03 (0.03) | | | | |
| **CHCl$_3$** | 0.13 (0.13) | 0.07 (0.04) | 0.06 (0.06) | | | | | |
| **CHBr$_2$Cl** | 0.75 (0.76) | 0.72 (0.56) | | | | | | |
| **CHBr$_3$** | 0.62 (0.48) | | | | | | | |

*Correlations for CH$_3$Cl over VEG sites were excluded one CH$_3$Cl measurement in TMRX−ET-1 (see Sect. 2.1.2).
Table 6 records the correlations between the measured VHOC fluxes, separately for all sites,
bare soil sites, vegetated sites (VEG), TMRX−ET sites and WM−KLY sites, as well as for the
sites closer to the seawater, including all COAST−TKM and COAST−EGD sites. For the two
last, correlations are also presented separately for the two SD sites closest to the seawater
(COAST−TKM-SD and COAST−EGD-SD). Note that the table compares net flux rather than
emission flux, and therefore the reported correlations are expected to be affected by both sinks
and sources for the different VHOCs.
Table 6 demonstrates moderate to high positive correlations in most cases when all sites are
included in the calculation, while in general the correlations were significantly higher when
calculated for sites of the same type, suggesting common emission mechanisms or controls. In
most cases correlations for the vegetated sites were higher than the overall correlations for all
sites. The relatively high correlations in the vegetated sites may be in line with previous studies
indicating high emissions from vegetation at marsh coasts (Rhew et al., 2002;Deventer et al.,
2018), but positive fluxes for methyl halides were obtained only in a few cases at the vegetated
sites, and not in all cases for all methyl halides simultaneously. Hence, it appears that the



correlations between methyl halides at the vegetated sites are more likely to be attributable to
common sinks. The fairly elevated correlations between methyl halide fluxes at the SD sites,
together with the fact that, in most cases, fluxes of the three methyl halides from these sites were
positive and high, suggests that these sites have a common source or sources for methyl halides.
High correlations were obtained for trihalomethanes in the vegetated sites ($r \geq 0.82$), except
for $CHCl_3$, whose correlations with other trihalomethanes were lower. We also observed high
correlations of $CH_2Br_2$ with all trihalomethanes, particularly for the vegetated sites ($r \geq 0.77$),
and somewhat lower correlations with $CHCl_3$ ($r = 0.55$). $CH_2Br_2$ also showed high correlations
with $CHBrCl_2$ ($r = 0.90$) and $CHBr_3$ ($r = 0.88$) at the SD sites, suggesting a common emission
mechanism for $CH_2Br_2$ and the other trihalomethanes.
Correlation of $CH_2Br_2$ with $CHBr_2Cl$ at the SD sites was strongly negative ($r = -0.93$),
similarly to the negative correlation between $CHBr_2Cl$ and the other brominated
trihalomethanes, $CHBCl_2$ ($r = -0.98$) and $CHBr_3$ ($r = -0.65$), at these sites. This, together with
the fact that the measured fluxes of these species were generally positive over the SD sites,
points to competitive emission between $CHBr_2Cl$ and both $CHBrCl_2$ and $CHBr_3$, at least at the
SD sites. This is supported by the analysis in Sects. 3.2.2 and 3.2.3, which have demonstrated
that the halide content of the soil appears to play a major role in controlling the emission rates of
VHOCs under the studied conditions. High positive correlation between all four brominated
species was observed for the bare soil sites as well as for the vegetated sites (see Table 6),
further supporting the notion that $CHBr_2Cl$ too can be emitted via mechanisms similar to those
of the other two brominated trihalomethanes and $CH_2Br_2$.
Table 6 also indicates overall low correlations between $CHCl_3$ and all of the brominated
trihalomethanes, mostly resulting from negative correlations at the bare soil sites. There was
also a higher incidence of positive fluxes at the bare soil sites for the trihalomethanes $CHCl_3$ and
$CHBrCl_2$, compared with the less chlorinated, $CHBr_3$ and $CHBr_2Cl$ (Table 4). Hence, the
negative correlation between $CHCl_3$ and the brominated trihalomethanes at the bare soil sites





may indicate competitive emission between the more chlorinated and the more brominated

trihalomethanes. The situation at the bare soil sites resembles previous reports of the

predominant emission of $CHCl_3$ at the expense of the more brominated species (e.g., (Albers et

al., 2017;Huber et al., 2009)), particularly $CHBr_3$ and $CHBr_2Cl$. This would be expected, given

the higher Cl/Br ratio at these sites (see Table 3). We should emphasize that even at the bare soil

sites we observed relatively high positive fluxes of brominated trihalomethanes, which would

not generally be expected (Albers et al., 2017), and can be attributed to the relatively high

bromine enrichment of the soil.

Interestingly, in agreement with Table 5, Table 6 also shows relatively high correlations

between $CHCl_3$ and all methyl halides, particularly for BARE ($r \geq 0.68$) and at the SD sites ($r \geq$

0.59). We also found high correlations for the SD sites between $C_2HCl_3$ and all methyl halides ($r$

$\geq 0.59$). Remarkably high correlations were obtained between $CH_3I$ and the brominated

trihalomethanes and $CH_2Br_2$ at the vegetated sites ($r \geq 0.57$), and for $CH_3I$ with $CHCl_3$ and

$C_2HCl_3$ at the SD sites ($r = 0.99$ in both cases). In most cases the flux of $CH_3I$ was negative at

the vegetated sites; therefore, it is not clear whether the strong correlations between $CH_3I$ and

the brominated trihalomethanes at these sites point to common sources or sinks. In contrast,

positive fluxes of both $CH_3I$ and the brominated trhihalomethanes and $CH_2Br_2$ were observed at

the SD sites in most cases, pointing to a common source of these species at the SD sites.

Weissflog et al. (2005) found that emission of $C_2HCl_3$, $CHCl_3$ and other chlorinated VHOCs can

occur from salt lakes via the activity of halobacteria in the presence of dissolved Fe (III) and

crystallized NaCl. The strong correlations of $CHCl_3$, $C_2HCl_3$ and $CH_3I$ ($r = 0.99$ in all cases)

reinforce the common emission of $CHCl_3$ and $C_2HCl_3$ from salt lake sediments, as indicated by

Weissflog et al. (2005), and may also indicate that $CH_3I$ can be emitted in a similar way. The

fact that the emission of $CH_3I$ in our study was much more efficient than under the conditions

used by Keppler et al. (2000) supports the possibility that mechanisms other than the abiotic

emission pathway proposed by Keppler et al. (2000) influence the emission of $CH_3I$ at the Dead





Sea (Sect. 3.2.3). The relatively high correlations between fluxes of $CHCl_3$ and $C_2HCl_3$ and the
other methyl halides, $CH_3Br$ and $CH_3Cl$, for Bare and SD, may suggest that these methyl halides
are also emitted, via similar mechanisms, from the salt deposits.

**Table 6.** Correlations between the measured net flux of VHOCs. The table records the Pearson correlation
coefficient (r) for the measured net flux between each VHOC pair, calculated over all sites except SEA−KDM.

| | | $CHBrCl_2$ | $CHBr_3$ | $CHBr_2Cl$ | $CHCl_3$ | $CH_2Br_2$ | $C_2HCl_3$ | $CH_3Cl$ | $CH_3Br$ |
|---|---|---|---|---|---|---|---|---|---|
| $CH_3I$ | All (n = 20) | **0.34** | **0.13** | **-0.56** | **0.58** | **0.19** | **0.59** | **0.45** | **0.23** |
| | BARE (n = 6) | -0.54 | -0.85 | -0.78 | 0.68 | -0.32 | 0.54 | 0.73 | 0.77 |
| | COAST (n = 10) | 0.50 | 0.26 | -0.64 | 0.66 | 0.81 | 0.63 | 0.54 | 0.08 |
| | SD (n = 4) | (0.13) | (0.72) | (-0.05) | (0.99) | (0.35) | (0.99) | (0.90) | (0.69) |
| | VEG (n = 4) | 0.76 | 0.72 | 0.57 | 0.31 | 0.88 | 0.16 | 0.11 | 0.45 |
| $CH_3Br$ | All (n = 20) | **-0.08** | **0.39** | **0.22** | **0.20** | **-0.06** | **0.33** | **0.30** | |
| | BARE (n = 6) | -0.22 | -0.83 | -0.45 | 0.83 | -0.21 | 0.57 | 0.61 | |
| | COAST (n = 10) | -0.51 | 0.65 | 0.19 | 0.29 | -0.04 | 0.33 | -0.24 | |
| | SD (n = 4) | (-0.62) | (0.07) | (0.69) | (0.59) | (-0.40) | (0.59) | (0.69) | |
| | VEG (n = 4) | 0.67 | 0.87 | 0.47 | -0.57 | 0.36 | -0.76 | 0.94 | |
| $CH_3Cl$* | All (n = 19) | **0.27** | **0.05** | **0.00** | **-0.37** | **-0.15** | **0.54** | | |
| | BARE (n = 6) | -0.33 | -0.63 | -0.54 | 0.86 | 0.21 | 0.71 | | |
| | COAST (n = 10) | 0.58 | -0.09 | -0.16 | 0.69 | 0.14 | 0.66 | | |
| | SD (n = 4) | (0.07) | (0.45) | (0.08) | (0.91) | (0.12) | (0.86) | | |
| | VEG (n = 3) | 0.45 | 0.68 | 0.31 | -0.75 | 0.06 | -0.91 | | |
| $C_2HCl_3$ | All (n = 20) | **0.10** | **0.53** | **0.05** | **0.83** | **0.02** | | | |
| | Bare (n = 6) | -0.41 | -0.66 | -0.52 | 0.56 | -0.10 | | | |
| | COAST (n = 10) | 0.30 | 0.65 | -0.01 | 0.99 | 0.26 | | | |
| | SD (n = 4) | (0.26) | (0.81) | (-0.19) | (0.99) | (0.48) | | | |
| | VEG (n = 4) | -0.05 | -0.34 | 0.12 | 0.96 | 0.33 | | | |
| $CH_2Br_2$ | All (n = 20) | **0.62** | **0.36** | **-0.17** | **0.15** | | | | |
| | BARE (n = 6) | 0.77 | 0.58 | 0.68 | 0.08 | | | | |
| | COAST (n = 10) | 0.45 | 0.26 | -0.85 | 0.27 | | | | |
| | SD (n = 4) | (0.90) | (0.88) | (-0.93) | (0.45) | | | | |
| | VEG (n = 4) | 0.91 | 0.77 | 0.87 | 0.55 | | | | |
| $CHCl_3$ | All (n = 20) | **0.01** | **0.30** | **0.01** | | | | | |
| | BARE (n = 6) | -0.25 | -0.74 | -0.46 | | | | | |
| | COAST (n = 10) | 0.31 | 0.60 | -0.04 | | | | | |
| | SD (n = 4) | (0.27) | (0.77) | (-0.18) | | | | | |
| | VEG (n = 4) | 0.22 | -0.09 | 0.40 | | | | | |
| $CHBr_2Cl$ | All (n = 20) | **-0.11** | **0.16** | | | | | | |
| | Bare (n = 6) | 0.95 | 0.86 | | | | | | |
| | COAST (n = 10) | -0.22 | 0.11 | | | | | | |
| | SD (n = 4) | (-0.98) | (-0.65) | | | | | | |
| | VEG (n = 4) | 0.82 | 0.94 | | | | | | |
| $CHBr_3$ | All (n = 20) | **0.22** | | | | | | | |
| | BARE (n = 6) | 0.72 | | | | | | | |
| | COAST (n = 10) | -0.04 | | | | | | | |
| | SD (n = 4) | (0.65) | | | | | | | |
| | VEG (n = 4) | 0.95 | | | | | | | |

* Correlations for $CH_3Cl$ over VEG sites were excluded one $CH_3Cl$ measurement in TMRX−ET-1 (see Sect.

2.1.2).


**Summary**


The results of this study demonstrate high emission rates of the investigated methyl halides as
well as of brominated and chlorinated VSLS at the Dead Sea area, corresponding with mixing
ratios which in most cases are significantly higher than typical values in the MBL. Overall, our
measurements indicate a higher incidence (in 65−85 % of measurements) of positive fluxes of
brominated than of chlorinated VHOCs, except for $CHCl_3$, for which the incidence of positive
net fluxes was also relatively high (65 % of measurements). The high incidence of brominated
VHOCs can be attributed primarily to the relatively large amount of Br in the soil, rather than
the Br/Cl ratio. We did not detect any clear effect of meteorological parameters, emission from
the seawater, or season, other than — in agreement with Sive et al. (2007) — an apparently
higher emission of $CH_3I$ during spring than during winter. The four investigated site types, the
cultivated and natural vegetated, the bare soil and the coastal sites, are identified as potential net
sources for all VHOCs investigated, except for the emission of $CH_3I$ and $C_2HCl_3$ from the
vegetated sites. Hence, this study reveals strong emission of VHOCs over at least a few
kilometers from the Dead Sea. The fluxes, in general, were highly variable, showing changes
between sampling periods even for a specific species at a specific site.

Emissions were highest from the SD sites, where salinity is maximal, and which clearly

showed an increased incidence of positive flux with decreasing distance from the seawater,
pointing to the sensitivity of VHOC emission rates to salinity even at the hypersaline coastal
area of the Dead Sea. The measurements did not indicate either increased or reduced emissions
of VHOCs from the seawater itself. It was shown that emissions of VHOCs can occur from dry
soil under semi-arid conditions during summer, in agreement with findings from other
geographic locations that soil water does not seem to be a limiting factor in VHOC emission
(Kotte et al., 2012). Rain events appear to attenuate the emission rates of VHOCs at the Dead
Sea. Measurements at a bare soil site suggested a decrease in VHOC emission rates for 1−3 days



after a rain event, while the gradual increase in VHOC emission more than three days after the
rain event suggests that these VHOC emissions are, at least partially, biotic-induced.
Trihalomethanes, including $CHCl_3$, $CHBr_2Cl$, $CHBr_3$ and particularly $CHBrCl_2$, are
associated with the highest number of sites at which their flux was, on average, positive, while
$CHBr_3$, $CHBr_2Cl$ and $CHBrCl_2$ showed relatively high incidence of positive fluxes, with values
of 65 %, 80 % and 85 %, respectively. This finding, together with the relatively high
correlations observed between brominated trihalomethanes, points to common formation and
emission mechanisms of these brominated trihalomethanes, in line with previous studies. Our
analyses further suggest emission of $CH_2Br_2$ via mechanisms that are common to the
trihalomethanes. Correlation of the brominated trihalomethanes with $CHCl_3$ was lower. Whereas
Albers et al. (2017) suggested that $CHBr_3$ and $CHBr_2Cl$ are emitted from soil only in relatively
small amounts compared to $CHCl_3$, our results point to their higher emission via common
mechanisms with the other trihalomethanes. The overall average net flux of the trihalomethanes
decreased according to $CHBr_2Cl > CHBr_3 > CHBrCl_2 > CHCl_3$. The enhanced emission of
brominated trihalomethanes probably reflects the enrichment of the Dead Sea soil with bromine,
in line with findings by Hoekstra et al. (1998), who identified a higher natural emission of
$CHBr_3$ and $CHBr_2Cl$ rather than of $CHBrCl_2$ from the soil, following the soil's enrichment with
KBr.
We identified the SD sites as a probable source for all methyl halides, whereas vegetated
sites appear more likely to act as a net sink for these species. Comparing the proportion of Br
and Cl in the soil for the various sites with proportions of measured positive flux of $CH_3Br$ and
$CH_3Cl$ are in line with reports by Keppler et al. (2001) about emission of methyl halides via
abiotic oxidation of organic matter in the soil. Similar calculations in our study demonstrated
much higher efficiencies of $CH_3I$ emission than those reported by Keppler et al. (2000), pointing
to emission of $CH_3I$ via other mechanisms. The high correlation of $CH_3I$ emission with that of
$CHCl_3$ and $C_2HCl_3$, particularly at the SD sites, together with findings by Weissflog et al.




(2005), of various chlorinated VHOCs emission, including $CHCl_3$ and $C_2HCl_3$, from salt lake
sediments, suggests that the Dead Sea, particularly the SD, sites probably act as an emission
source for $CHCl_3$, $C_2HCl_3$ and $CH_3I$ via similar mechanisms. Weissflog et al. (2005) reported
that the emission of chlorinated VHOCs in their study was induced by microbial activity.
Keppler et al. (2000) reported the involvement of an abiotic process in the formation of alkyl
from soil and sediments, and the observed correlation between methyl halides and both $CHCl_3$
and $C_2HCl_3$ may indicate that the two processes occur simultaneously. Further research will be
needed to decipher the relative importance of each process in soil and salt sediments, including
more direct emission measurements from a better-defined landform, e.g., by using flux
chambers.
Of all the VHOCs investigated in our study, $CHBr_3$ showed the highest enrichment with
respect to MBL mixing ratios. Owing to the relatively short tropospheric lifetime of $CHBr_3$, its
photolysis contributes significantly to reactive bromine formation in the MBL. However,
although relatively high, the elevated $CHBr_3$ fluxes and mixing ratios that we measured at the
Dead Sea, cannot lead to the elevated mixing ratios of reactive bromine species at the Dead Sea,
which are frequently associated with BrO > 100 ppt (e.g., see Matveev et al. (2001) and Tas et
al. (2005)). Similarly, if $CH_3I$ photolysis is the only source of reactive iodine species, the
measured fluxes and elevated mixing ratios of $CH_3I$ are not high enough to account for the high
IO in this area. Given their relatively fast photolysis, however, $CH_3I$ and $CHBr_3$, as well as
$CH_2Br_2$, may well have roles to play in the initiation of reactive bromine and iodine formation in
this area.
Overall, along with other studies, the findings presented here highlight the potentially
important role played by emission of VHOCs from saline soil and salt lakes in stratospheric and
tropospheric chemistry, and call for further research on VHOC emission rates and controlling
mechanisms.



**Data availability**. Data are available upon request from the corresponding author Eran Tas (eran.tas@mail.huji.ac.il).

**Author** *contribution*: ET, AG and RR designed the experiments. MS, GL and QL carried the field measurements out and DB carried out the sampled air analysis. GL contributed in designing and constructing a special mechanism for simultaneous lifting and dropping of sampling canisters. Data curation and formal analysis were performed by ET and MS with support from RR. ET and MS and ET prepared the manuscript with contributions from all co-authors.

**Competing interests**. The authors declare that they have no conflict of interest**.**

**Acknowledgements**

This study was supported by United States−Israel Binational Science Foundation (Grant 2012287). E.T. holds the Joseph H. and Belle R. Braun Senior Lectureship in Agriculture.

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
