# Peer review of "Manuscript under review for journal Atmos. Chem. Phys."

_Atmospheric Chemistry and Physics, 2018_

## Author Comment (AC1) · 21 Dec 2018

There were errors in the authors' affiliations. The correct affiliations are:

Moshe Shechner [1], Alex Guenther [2], Robert Rhew [3], Asher Wishkerman [4], Qian Li [1], Donald Blake [5], Gil Lerner [1], and Eran Tas [1]

1. The Robert H. Smith Faculty of Agricultural, Food & Environment, Department of Soil and Water Sciences, The Hebrew University of Jerusalem, Rehovot, Israel; 2. Department of Earth System Science, University of California, Irvine, CA, USA; 3. Department of Geography, University of California, Berkeley, CA, USA; 4. School of marine

sciences, Ruppin Academic Center, Michmoret, Israel; 5. Department of Chemistry, University of California, Irvine, Irvine, CA, USA

I sincerely apologize for these errors, Eran
* * *

---

## Referee Comment (RC1) · Anonymous Referee #1 · 16 Jan 2019

This is a study of concentrations of short-lived halogenated gases from a unique area, the Dead Sea. Given the interest in these chemicals and the uniqueness of this location, this paper has the potential to be an interesting contribution. It certainly includes a thorough review of the available literature and the authors have very thoroughly considered their new results in light of previously published work. However, I'm concerned about a number of aspects of the interpretation of the measurements, which are described below. The most significant is an inadequate consideration of uncertainties in most aspects of the work. This leads to an extended discussion throughout the paper of effects that I'm not convinced are real.

[Figure]

In Table 2, comparisons are made between concentrations measured at these Dead Sea sites with reported concentrations in the marine boundary layer (MBL) (as medians, from Ozone Assessment Reports), and measured concentrations enhancements are taken to imply significant local emissions. But this seems an inappropriate conclusion. I would expect that the influence of the marine boundary layer on what is being measured in the Dead Sea valley is diminished by the time air moves from any distant sea (Red or Mediterranean) to this valley owing to vertical mixing within the lower atmosphere. Perhaps instead, any enhancement relative to the MBL suggests only that fluxes are non-zero in this region too, and are perhaps comparable (or larger) than suggested for the marine boundary layer and coastal ecosystems? Drawing conclusions from concentration differences in the Dead Sea area vs the MBL is tricky and not especially informative, given that concentrations are influenced by dynamics in addition to flux–this seems worth mentioning, but isn't yet in this regard. Also, why aren't MBL fluxes also shown in Figure 2?

I also find it very difficult to internalize the information given in Table 2 as presented. I'd recommend the presentation of these results, if retained, also (or instead) as a figure. Furthermore, I'd suggest that any enhancement factor should also consider the reported range in the marine boundary layer concentrations so that the reader can better understand the degree to which the Dead Sea region concentrations actually are anomalous (regardless of reason, flux or meteorology). In addition, for some of these gases there are some well documented temporal, seasonal, and latitudinal variations in MBL concentrations that aren't well considered by the "annual average for 2012". As a result, I suspect that some of the EF's ($CH_3Br$, perhaps also $CH_3Cl$, $CHCl_3$, and $CH_2Br_2$) are not accurate representations.

On fluxes, the text seems to inaccurately reflect what the figure indicates once uncertainties are considered. One example: "Figure 2 presents the measured fluxes of all VHOCs studied. On average, the net fluxes of all measured species, except $C_2HCl_3$ and $CH_3I$, were positive at most of the investigated sites", and my review of Figure

2 indicates a much lower occurrence of positive flux: only 13 of 36 panels (excluding CH3I and C2HCl3) show positive fluxes where the standard error does not encompass zero. Another example can be found in section 3.2.1, lines 417-419. It is necessary to consider the uncertainty on the average here in drawing conclusions. Furthermore, I would estimate that the standard errors are likely underestimated as a result of the fairly small number of measurements used to estimate fluxes in this work.

Are the fluxes actually associated with the ecosystems purportedly sampled by this technique? Inferences about flux from measurements as a function of height have a certain spatial influence function. Please indicate what that might be for the sampling heights you have chosen. Consideration of the C2HCl3 results (an implied sink, perhaps from elevated mixing ratios in the broader Red Sea region) may indicate that the fluxes you are deriving here for naturally-emitted gases are actually not representative of the local regions you intended them to represent. How is the reader to assess this? Also, what has determined the different heights at which samples were collected on these masts? Sampling heights in a region with local emissions should have a large, but not discussed, impact on measured mixing ratios—which are being compared among sites and to MBL results.

I find the results in Figure 3 intriguing, although not much is made of it in the text. While it may be that no generalizations are possible related to all gases, there are some interesting similarities that might be worth discussing, especially to understand if these co-variations are consistent with the discussions related to co-variations in fluxes as what was intended in Table 4. Table 4 is also very hard to extract information from... and as before, I'm concerned that any identification of positive flux amounts don't take into account uncertainties on those estimations. If uncertainties were not considered, then it seems that much of the discussion related to incidences of positive flux and rankings by chemical etc. that follows should be reconsidered.

Table 5 and 6 need a consideration of correlations that are and are not significant, given the number of measurements included in each determination. Given the small

number of samples considered here, I'd estimate that correlations of <0.1 are in fact indicative of no evidence for a correlation, not a correlation described as "low".

---

## Short Comment (SC1) · 3 Feb 2019

I want to greatly thank the reviewer for the constructive and helpful comments, and address here the main concern raised by the reviewer, regarding the uncertainties of the measured fluxes. We agree that uncertainties should be taken into account in the flux analyses, and therefore we applied a Student's t-test to test if the fluxes are significantly different from zero. Taking into account the results of this test in our analyses, did not lead to significant changes in the conclusions. Some assumptions that were made based on the correlation analyses (Tables 5 and 6) should be indeed revised due to the small number of measurements. We will address all comments in

detail, including uncertainties in the mixing ratios, prior to submission of the revised manuscript. Sincerely, Eran

---

## Referee Comment (RC2) · Anonymous Referee #2 · 22 Feb 2019

**General Comments**

The manuscript by Shechner et al. presents ambient measurements and fluxes for short-lived halocarbons at multiple sites around the Dead Sea. The unique characteristics of the Dead Sea make it a very interesting location to study the emissions from and detail the characteristics of this source for atmospherically important halocarbons. The paper contains an abundance of information, but I feel some key details are lacking that are needed to fully assess the author's interpretations. Additionally, the paper is very long and becomes difficult to follow in terms of the main points trying to be conveyed in the various sections of the paper. My opinion is that the paper could be distilled down in length and the key points be fleshed out a bit more cleanly. Additionally, I feel there are some significant improvements that could be made in dissemination of the information in both graphical and tabular form. While there is merit to the manuscript, I feel as though there are an array of issues that should be addressed before it is in an acceptable format for publication.

I will present a general list of issues here and elaborate on them in the Specific Comments section.

Urban and other source influences – it would be useful to provide some context to the potential of urban emissions, for the solvents like CHCl3 and C2HCl3, but also including things like wastewater treatment facilities and other agricultural activities that could influence the area.

The first time a chemical constituent is introduced, it should be spelled out – there are several places this occurs throughout the manuscript. For example, L72 chloroform (CHCl3), L73 chloroethane (C2H5Cl), L112 iron (Fe), L115 potassium bromide (KBr), nitric acid (HNO3), etc. – please address.

Percentages – there are spaces between the number and the percent sign. The most common convention is to not have a space between a number and the percent sign.

I would recommend either referring to the suite of halocarbons as VHOCs or VSLS, but not going back and forth between them.

From the measured fluxes, can you estimate the local/regional source or sink strength of the Dead Sea? How do your results play in to the scale of the source strength of the Dead Sea for these gases?

It would be useful to present some quantitative information in the abstract, such as mixing ratios and fluxes.

There are no uncertainties propagated through any of the fluxes.

I would be useful to include the atmospheric lifetimes and primary removal sources for the compounds in the manuscript.

The manuscript seem to try and agree with all previous studies.

Tables are difficult to read and digest.

Plots within the figures are too small making it difficult to extract information from them.

Flux section could be moved to SI

A more thorough overview of the site, including meteorology, would be useful to help set the stage for the reader.

**Specific Comments**

L46-7:  You should include why CH3I and C2HCl3 are exceptions, as this is not intuitive to the reader.

L49-51:  For the statement:  "Correlation analysis, in agreement with recent studies, indicated common controls for the formation and emission of all the above trihalomethanes but also for CH2Br2.", I'm not convinced this is entirely accurate – for example, what about CHCl3?  Also how does the correlation indicate that the factors controlling the formation and emissions are the same?

L55: "elevate" should be "elevated"

L61:  When you introduce VSLSs here, you should include here that this refers to compounds that have lifetimes of less than 6 months.

L64:  replace "destruction of ozone" with "ozone destruction"

L73:  add "which", so it reads "…C2H5Cl, which originate…"

L134-5:  Bromide (Br-) and chloride (Cl-) should be introduced and the sentence should be revised to read " with water salinity 12 times higher and a bromide to chloride ratio (Br−/Cl−) 7.5 times higher than in normal ocean waters.
L136: What do you mean by "landforms"?  Formations from the residual salts left behind?  In this case the use of the term "landform" invokes images of large scale topographical features, is this the case?

This brings in to question the use of the term landform in the title – is this really appropriate and accurate?  I would say this work has been carried out on different terrains or ecosystems of the Dead Sea, but not different landforms.

L143:  I would revise this to make it a stronger statement, something like:  "Studying the emission of VHOCs at the Dead Sea is also fundamental for understanding local surface ozone depletion events…"

L169-70:  Regarding the Tamarix vegetation and watermelon fields, more details, such as density, proximity, size of agricultural development, etc., would be useful to the reader.

Also, I would refer change your referencing of watermelon fields from "vegetation" to "agriculture" in later sections of the manuscript – because this is a perturbed system different that the natural vegetation, it should be distinguished as such.

L198:  Revise to "Lastly, WM-KLY…"

**P8, Sect. 2.1.2.**

How many samples were collected in total, at each site, and at each corresponding height for each site? What were the meteorological conditions during the sampling?

In order to get better feel for the results presented, both for the ambient levels and the fluxes, knowing *N* is critically important. This will allow the reader better perspective on some of the interpretation presented.

Also, general information about the seasonal and local meteorology to provide an overview of the region would be instructive to the reader.

L209: Regarding the use of "fast" here: I personally wouldn't consider 20 minutes to be fast - I think the key point you are trying to make is that all samples were collected simultaneously and integrated over a 20 min period – please revise.

Also, I'm assuming "lifting of the canisters" should be "filling of canisters"

L209-12: Please revise the following sentence – very awkward as written:

Facilitated by passive grab samplers (RESTEK Corporation, PA, U.S.), we performed each sampling within 20 minutes by pulling air into evacuated 1.9 L stainless steel canisters, resulting in an internal canister pressure higher than 600 torr.

L215-16: Please revise: "…subjected to the analytical techniques…" – simply say they were analyzed by similar techniques described in Colman et al.

L218-21: You introduce all of the halocarbons here, but most, if not all should have been introduced previously. Please address.

L223-28: Please provide some statistical/quantitative rationale for this - you can't simply disregard this point because it doesn't "agree" with the other measured mixing ratios for CH3Cl. Also, I don't feel it's appropriate to state that it may result in a "less accurate flux" – how do we know what the "accurate flux" is? There is variability in all of this work, and while this may, in fact, be a spurious data point, what measures were carried out to deduce this issue?

Where is this listed, in Table 2? Please specify here for the reader to address.

**L229, Table 1:** It would be useful to provide the total number of samples and how many per height. This should be summarized such that the reader doesn't have to try and count how many samples were collected on the individual days from the information in the table.

**L279: Sect 2.3**
Following suit with the canisters, how many total soil samples were collected and analyzed? This potentially could be moved to the SI because the information os only used for general properties at each site.

L280-81: Please elaborate what you mean by this and what is the significance of this statement: "…at least 3 months following any rain event in the Dead Sea area."

L265: In line reference should be Golder (1972)

L290: Quotes are not needed around "Discover"

L292-93:  High Resolution does not need to be capitalized

L294:  "low-limit" should be "lower limit"

L302:  What is meant by "corresponding available information."

L306:  What is the "Dead Sea Works"?

L326:  There were surface seawater, ambient air and direct flux measurements of CHBr3 in Zhou et al., 2005 – how do these compare with the Dead Sea?

L330:  It appears that a range of values is missing after 2-60 pptv – there is simple "(-)"

L335:  Re C2HCl3 and CH3I - while the reader can look at the figure, it would be useful to also state in the text what these gases are doing, on average.

L336-39:  Can you please clarify these two sentences: Figure 2 doesn't show values higher than these.  Either present the values or revise text.

L376: for the following, you either have one too many or one too few brackets:  (e.g., ~600 nmol m$^{-2}$ d$^{-1}$; (Deventer et al., 2018).

L360:  For "nmol m$^{-2}$d$^{-1}$" there appears to be an extra dash in between m$^{-2}$ and d$^{-1}$

L391-93:  It is difficult to see this in Fig 2, and what/where are the anthropogenic emissions located?  From DSW or other places?  Can this be assessed by looking at something like the C2HCl3/C2Cl4 ratio?  It is likely that this data is available from the UCI group, but this (or other pairs of compounds) could be used to do a more thorough analysis on the impact of anthropogenic emissions at the sampling sites.  For example, this brings in to light things like wastewater treatment facilities and the corresponding emissions of CHCl3 and CHBr3.  It would be useful to provide a more rigorous assessment of the influence of anthropogenic emissions in general - particularly for those not familiar with the region and to what extent they may be influencing this work - if minimal, that's great - just demonstrate this, as this statement affects your results - C2HCl3 isn't the only gas here with anthropogenic sources.

Suggestion:  After looking at Figure 2, I feel as though it would be useful to have a summary flux figure (e.g., by compound) with the magnitude of the fluxes plotted by size or color on a map to enable the reader to get a better idea of the spatial variability of the flux magnitudes.

L418:  Revise to: "…VHOCs, except C2HCl3, were…"

L431-33:  Regarding the statement that there isn't a difference between fluxes in the spring and winter, two things should be addressed:  1) is this statistically significant?  2)  What is the seasonality of the temperature and overall meteorology for this area (i.e., local/regional transport patterns)?  Being only slightly extratropical, would seasonality be expected to be an important driver?

L437:  add comma after "properties"

L439-40: Regarding the sentence: "No clear impact of season or distance from the seawater on the mixing ratios can be discerned in this figure,…", while I agree, it's mostly because you can't see the details in Figure 3.

Figure 3: In general, it is difficult to discern the spatial distributions and get useful information out of the vertical profiles because each panel is so small. From this figure, it is difficult to see and discern the gradients for many of the gases. I would recommend revising and either show a few key species and put the remainder that don't show anything in the SI or revise the whole figure.

L460: replace "these parameters" with "the soil composition parameters"; also change "The table records…" to "The results presented in Table 3 show…"

L462: "larger distance" should be replaced with " greater distances"

L465-66: replace "in" with "at" just before the site location abbreviation.

L472-75: What do you mean by "underestimated value of Fe"? A lower limit of the total iron? Again, "low-limit" should be "lower limit".

What does "while the emission rates became saturated" mean – I'm assuming that you mean "plateau". For example, Huber et al. use the term "plateau".

L478: I would replace "merges" with "combines"

L481: "samplings" should be "sample", and I would encourage revising this sentence to something like: "While the number of samples collected at each site was limited, Table 4 shows that the fluxes...."

L482: I would replace "In both" with "For the…sites,…"

L484: replace "in" with "at" before COAST-EGD-MD

L485: A comma is needed after "winter"

L505: For consistency, replace VSLS with VHOC.

L506-07: Replace "during" with "at" before the site abbreviations.

L515-19: Do you need the F:C ratio really aid in understanding these processes?

L544: Table 4. General comment: This is a hard table to read and extract information from - I almost feel as though presenting this graphically would be more impactful allowing the reader to see the trends rather than sifting through a lot of numbers that appear to vary greatly.

Because everything is bolded in the summary portion of Table 4, the rows should be explicitly labeled as to what the values are.

L553: Awkward as written, say something like: The results presented in Table 4 show that a higher…"

L554: comma is needed after CHCl3

L555: "tends" should be "tended"

L558: I would suggest deleting the following (not needed): "suggesting both high emission and their balance to some extent by sinks for this species."

Because there were watermelon fields, was there any harvesting or drying and decomposing plant material in the vicinity of the sampling? This can be a source of an array of halocarbons, particularly gases like CHCl3 and CHClBr2.

L563: I would replace "are in general" with "were"

L566-70: Please revise – it is unclear what you are trying to say.

L577-9: revise to something like: "…emission rates from both bare and vegetated soil sites supports the work by Albers et al. (2017) concerning the emission of trihalomethanes from the soil after trihaloacetyl hydrolysis (Table 3)."

L584: Agricultural emissions, such as from the watermelon farming, could be such a source. More details regarding the scale and influence of these operations would be useful.

L589: replace "in" with "at" before the site name

L589-91: Please revise the following – awkward as written: "No clearly more elevated positive flux of brominated compared to chlorinated trihalomethanes was observed for this site…"

L600: include (Table 4) to direct the reader to this information

L601: For the statement "…indicating strong emission and deposition…", if the flux is positive, then the emissions outweigh the deposition or other loss processes - revise to clarify your point. Figure 2 counters the point of "strong deposition" for the methyl halides.

L604-05: Cultivated watermelon fields (agricultural emissions) are different from local vegetation, please distinguish as such.

L644-46: please revise, reads awkwardly

L662-663: How are the data grouped for Table 5? Is this simply for all sampling heights lumped together? Is there a difference when grouped by height?

Replace "evaluated" with "measured"

L670: Please consider revising: "…reinforce predominant contribution of VHOCs from terrestrial sources…" – I would consider this to be an overstatement.

L672: The $r^2$ values are quite low, and without being able to see the correlation plots of these gases, it is difficult to adequately assess the commonality of their sources and sinks. How do these specific $r^2$ values translate in to common sources and sinks?

L678: Replace "records" with something like "shows"

L680-81:  Change "For the two last,…" to something like: "For the latter two sites,…"

L685:  Replace "demonstrates" with something like "shows" or The results in Table 6 show/illustrate…

L692-94:  Can you please expand upon the correlations being attributable to "common sinks" – what are the sinks and how is this driving the correlations?

L740:  replace "common emission" with something like "co-located emissions"

L826-27:  I would recommend revising or omitting the following:  "…from saline soil and salt lakes in stratospheric and tropospheric chemistry,…", as there were no linkages made to how the compounds measured for this work play in to the local/regional/global budgets of tropospheric or stratospheric Cl, Br or I.

---

## Author Response (AR2)

Dear Editor,

We are pleased to submit the revised version of the manuscript (acp-2018-1172)

"**Emission of volatile halogenated organic compounds over various Dead Sea landscapes**".

      First we want to deeply thank the two reviewers for the effort they invested in reviewing this paper and for its thorough and constructive review. The review helped us better support our findings, improve the presentation of the results and give a more complete, clear and concise discussion. We seriously considered all of the reviewers' comments. We hope that our important scientific findings will be found acceptable, following the revisions that we have made to the manuscript, as described in the following. We open this response with a general description of the major revisions in the manuscript, followed by detailed point-by-point responses to each of the reviewers' comments.

Sincerely,

      Eran Tas

**General major revisions according to the reviewers' comments**

- All discussions and conclusions are supported by statistical tests.
- Title: We now use "landscapes" instead of "landforms" in the title.
- Abstract: We report the mixing ratios and flux ranges for all investigated volatile halogenated organic compounds (VHOCs).
- We present a figure comparing the measured mixing ratios of all investigated VHOCs instead of the original Table 2. A revised version of this table is included in the Supplementary Information (Table S3).
- Fluxes in the original Table 4 (now Table 2) as well as the related discussions are now reported along with information about their statistical significance. Table 2 has been moved to Sect. 3.1.

- Fluxes in Fig. 3 (originally Fig. 2) as well as the related discussions are now reported along with information about their statistical significance. The figure has been revised to allow easier information extraction.

- A new figure has been added (Fig. 4) to demonstrate the spatial distribution of the VHOC fluxes in the studied area.

- Fig. 5 (originally Fig. 3) has been revised and the original figure is included in the Supplement (Fig. S1).

- Correlation values between mixing ratios in Table 4 (originally Table 5) are reported along with their corresponding statistical significance. These values are now also provided individually for different site types (bare soil, coast, etc.).

- Correlation values between fluxes in Table 5 (originally Table 6) are reported along with their corresponding statistical significance.

- The following has been added to the Supplement: average lifetime and primary removal pathways for the VHOCs (Table S1); a tabulated comparison of the mixing ratios measured in this study and the corresponding values reported for the marine boundary layer (revised original Table 2 (now Table S3)); all sampling footprints (Table S4); analysis of potential anthropogenic influence during the measurements (Sect. S5; also discussed in the main text); in situ measured meteorological parameters during the air sampling (Sect. S6).

In the following, all of the reviewers' comments (in italic red font) are followed by our detailed responses.

**Response to comments by reviewer #1**
**1.** *This is a study of concentrations of short-lived halogenated gases from a unique area, the Dead Sea. Given the interest in these chemicals and the uniqueness of this location, this paper has the potential to be an interesting contribution. It certainly includes a thorough review of the available literature and the authors have very thoroughly considered their new results in light of previously published work. However, I'm concerned about a number of aspects of the interpretation of the measurements, which are described below. The most significant is an inadequate consideration of uncertainties in most aspects of the work. This leads to an extended discussion throughout the paper of effects that I'm not convinced are real.*

*In Table 2, comparisons are made between concentrations measured at these Dead Sea sites with reported concentrations in the marine boundary layer (MBL) (as medians, from Ozone Assessment Reports), and measured concentrations enhancements are taken to imply significant local emissions. But this seems an inappropriate conclusion. I would expect that the influence of the marine boundary layer on what is being measured in the Dead Sea valley is diminished by the time air moves from any distant sea (Red or Mediterranean) to this valley owing to vertical mixing within the lower atmosphere. Perhaps instead, any enhancement relative to the MBL suggests only that fluxes are non-zero in this region too, and are perhaps comparable (or larger) than suggested for the marine boundary layer and coastal ecosystems? Drawing conclusions from concentration differences in the Dead Sea area vs the MBL is tricky and not especially informative, given that concentrations are influenced by dynamics in addition to flux–this seems worth mentioning, but isn't yet in this regard. Also, why aren't MBL fluxes also shown in Figure 2?*

**Answer**: Thank you for this comment. The only reason that we compared concentrations at the Dead Sea to those of the MBL was to suggest irregularly high local emissions in the Dead Sea area, rather than from either the Red Sea or the Mediterranean Sea, whose contributions to the local concentrations are indeed expected to be negligible. Considering the relatively large distance from the Mediterranean Sea (~90 km) and the Red Sea (160 km), we believe that these elevated concentrations imply local emissions from the Dead Sea area itself. We did not intend to indicate significant contributions from the Red or Mediterranean seas, and to clarify this, we now state the following: "Overall, the measurements at the Dead Sea boundary layer revealed higher mixing ratios for all investigated VHOCs than their expected levels at the Mediterranean Sea and Red Sea MBL, indicating higher local emissions from the Dead Sea area" (lines 333-335). We agree that comparing the mixing ratios at the Dead Sea to those measured at the MBL is tricky, but we think that it provides some understanding of how this area can contribute to VHOC loading relative to nearby marine environments. Taking this and the next comment into account, we now present the comparison of mixing ratios to those in the MBL in a figure without including enrichment factors.

We do not include MBL fluxes in Fig. 3 (originally Fig. 2) because reported fluxes in the MBL were measured under very different conditions, which also resulted in remarkably large differences in their magnitudes. For instance, fluxes of VSLSs have been found to be significantly different in magnitude over the coastal area, open ocean, shelf and upwelling (Carpenter et al., 2009). Moreover, we also compare the measured fluxes with those measured over various landscapes, such as bare soil, soil mixed with salt deposits, and vegetation, because in the case of fluxes, we find it more suitable to compare the reported fluxes more selectively in the text.

**2.** *I also find it very difficult to internalize the information given in Table 2 as presented. I'd recommend the presentation of these results, if retained, also (or instead) as a figure. Furthermore, I'd suggest that any enhancement factor should also consider the reported range in the marine boundary layer concentrations so that the reader can better understand the degree to which the Dead Sea region concentrations actually are anomalous (regardless of reason, flux or meteorology). In addition, for some of these gases there are some well documented temporal, seasonal, and latitudinal variations in MBL concentrations that aren't well considered by the "annual average for 2012". As a result, I suspect that some of the EF's (CH3Br, perhaps also CH3Cl, CHCl3, and CH2Br2) are not accurate representations.*

**Answer**: We agree, and the results presented in former Table 2 are now included as Figure 2 in the new manuscript. Table 2 is now presented in the Supplement, but instead of showing the enrichment factors, we explicitly show the reported measurements for the MBL as compiled by Carpenter et al. (2014). We agree that presenting enrichment factors is problematic because of the sensitivity of the concentrations to season, latitude, meteorological conditions, investigated area within the MBL, etc., particularly for the specified species ($CH_3Br$, etc.). Therefore, we do not include emission factors in the new version, and we think that presenting the results vs. the information compiled for the MBL in a graphical way, including a range for the MBL mixing ratios, is a reasonable way to compare the two data sets. Factors which may lead to biased comparison between mixing ratios measured at the Dead Sea and those measured in the MBL are now discussed: "It should be noted, however, that while Fig. 2 implies elevated VHOC emission from the Dead Sea, comparison of mean or median mixing ratios of VHOCs for the Dead Sea with those for the MBL is not straightforward, considering that VHOC mixing ratios in the MBL are sensitive to several factors, including season and latitude. Moreover, the measurement height can play a significant role in affecting the mixing ratios due to decreasing mixing ratios with height over areas where local emissions occur. Hence, we also compared the measured fluxes and mixing ratios with their corresponding values measured in coastal areas, where the highest mixing ratios in the MBL were generally measured due to stronger emissions." (lines 347-355). We also refer specifically to differences in sampling heights with respect to Fig. 5: "Note that differences in sampling heights at different sites can lead to a biased comparison between mixing ratios at different sites; nevertheless, in most cases, differences across measurement sites were larger than across vertical heights. " (lines 592-594).

**3.** *On fluxes, the text seems to inaccurately reflect what the figure indicates once uncertainties are considered. One example: "Figure 2 presents the measured fluxes of all VHOCs studied. On average, the net fluxes of all measured species, except C2HCl3 and CH3I, were positive at most of the investigated sites", and my review of Figure 2 indicates a much lower occurrence of positive flux: only 13 of 36 panels (excluding CH3I and C2HCl3) show positive fluxes where the standard error does not encompass zero. Another example can be found in section 3.2.1, lines 417-419. It is necessary to consider the uncertainty on the average here in drawing conclusions. Furthermore,I would estimate that the standard errors are likely underestimated as a result of the fairly small number of measurements used to estimate fluxes in this work.*

**Answer**: We have addressed this comment by rigorously taking statistical significance into account throughout all of the analyses and discussions, and in drawing the related conclusions. Fluxes in Table 2 (originally Table 4) are presented along with $p$-values that indicate their statistical significance, for a specific species at a specific measurement site, by applying a one-sample t-test. Note that considering the small number of measurements, these $p$-values are presented in four different categories: $p < 0.05$, $0.05 < p < 0.1$, $0.1 < p < 0.15$ and $p > 0.15$. For our analyses, only $p$-values $<0.05$ are considered, indicating that a specific site is a net source or sink for a specific species, while the other two $p$-value categories (excluding $p > 0.15$) are used only to indicate a moderate likelihood of the fluxes being either positive or negative, possibly due to the small number of measurements. Note that in several cases, correlation analyses of both flux and mixing ratios strongly support the emission of species from a specific site, although the corresponding flux is reported as insignificant in Table 2; for instance, remarkably high correlations were found for $CH_3I$ with $CHCl_3$ and $C_2HCl_3$ at the coastal sites near the seawater ($r = 0.99$, $p < 0.05$ in both cases), based on the flux correlation analysis (Sect. 3.3, lines 883-887), also supported by the concentration correlation analysis (Sect. 3 lines 887-895). Nevertheless, these coastal sites were found insignificant as a net source for $CHCl_3$ ($0.05 < p < 0.10$), and for this reason we think that presenting also $0.10 < p$-values $< 0.15$ and $0.05 < p$-values $< 0.10$ contributes, particularly for future studies in this field, even though these are not taken into account by our analyses.

Note that we also tested the $p$-value calculations assuming that EGD-SD and TKM-SD, as well as BARE–MSMR and BARE–MSD, are the same emission source, considering their similar characteristics (see Sect. 3.1). In the case of the SD sites, this assumption resulted in a lower evaluated $p$-value in only a few cases, affecting the $p$-value category (ranking; see Table 2). The statistical tests related to the information presented in Table 2 are described in its caption (see lines 495-497). This table was moved to Sect. 3.1 to support the reports on measured fluxes at the Dead Sea, based on the statistical analysis that is incorporated in the table. We have extensively changed the text in Sect. 3.1 and in Sect. 3.2.1–3.2.3 as a result of taking into account the uncertainties associated with the measured fluxes, based on the statistical analysis. In addition, the two specific comments given as an example by the reviewer were addressed (lines 419-421 and lines 549-552). Overall, however, the conclusions drawn based on the fluxes that are presented in Table 2 were not changed significantly. This is mainly because there is a general correlation between the incidence of positive fluxes associated with $p$-values $\leq 0.05$ and that of the fluxes which were considered positive in the original version of the manuscript. We have double-checked the standard errors, provided by us as standard mean errors, and we find them to be accurate.

Regarding the correlation analyses, each correlation value (i.e., Pearson correlation coefficient ($r$)) in Tables 5 and 4, (originally Tables 6 and 5, respectively) is reported along with the corresponding $p$-value to indicate whether the correlation is significantly different from zero, based on Student's t-test. In the case of correlation coefficients, $p$-values are also reported in four different categories: $p < 0.05$, $p < 0.1$, $p < 0.15$ and $p > 0.15$. For the analyses, only $p$-values $<0.05$ were considered, indicating that a specific correlation is significant or not significant, respectively, while the other two $p$-value categories (excluding $p > 0.15$) were used only to indicate moderate likelihood of the fluxes being either positive or negative, taking into account the small number of available measurements.

The revisions resulting from taking into account the uncertainties for the analyses are included in sections 3.1–3.3, and in the following, we present the related revisions in the summary section, showing their overall effect on our findings and conclusions.

1. In the original version: "Overall, our measurements indicate a higher incidence (in 65−85 % of measurements) of positive fluxes of brominated than of chlorinated VHOCs, except for $CHCl_3$, for which the incidence of positive net fluxes was also relatively high (65 % of measurements)." (lines 756-759).

In the revised version: "Overall, our measurements indicate a generally elevated incidence of positive fluxes of brominated vs. chlorinated VHOCs compared to previous studies" (lines 910-911). Hence, this statement is valid based on the flux uncertainties, namely, considering a measured flux as positive only if the related measurement site was identified as a statistically significant ($p < 0.05$) source for the specific species (for which the flux was measured). In the revised version we do not specify the percentage of this positive flux, but the reader can find this information in Table 2, in several different statistical significance categories. We have removed the text on the incidence of $CHCl_3$ considering the relatively low incidence of positive flux from sites which were identified as a source for $CHCl_3$ ($p < 0.05$).

2. In the original version it was mentioned that: "The four investigated site types, the cultivated and natural vegetated, the bare soil and the coastal sites, are identified as potential net sources for all VHOCs investigated, except for the emission of $CH_3I$ and $C_2HCl_3$ from the vegetated sites. Hence, this study reveals strong emission of VHOCs over at least a few kilometers from the Dead Sea" (lines 763-767).

In the revised version: "Three of the investigated site types – bare soil, coast and agricultural field – were identified as statistically significant ($p < 0.05$) sources for at least some of the investigated VHOCs. The fluxes, in general, were highly variable, showing changes between sampling periods, even for a specific species at a specific site. The coastal sites, particularly at a short distance from the sea (SD sites) where soil is mixed with salt deposits, were sources for all of the investigated VHOCs, but not statistically significantly for $CHCl_3$. Further from the coastal area, the bare soil sites were sources for $CHBrCl_2$, $CHBr_2Cl$, $CHCl_3$, and apparently also for $CH_2Br_2$ and $CH_3I$, and the agricultural vegetation site was a source for $CHBr_3$, $CHBr_2Cl$ and $CHBrCl_2$. Our measurements reinforce reports of $CHCl_3$ and $CHBrCl_2$ emission from bare soil, but indicate that such emission can also occur under relatively low soil organic content. To the best of our knowledge, we report here for the first time strong emission of $CHBr_2Cl$ and emission of $CH_2Br_2$ from hypersaline bare soil, at least a few kilometers from the Dead Sea. We could not identify the contribution of either natural or agricultural vegetation to the emission of the investigated VHOCs." (lines 915-928).

3. In the original version: "Measurements at a bare soil site suggested a decrease in VHOC emission rates for 1−3 days after a rain event, while the gradual increase in VHOC emission more than three days after the rain event suggests that these VHOC emissions are, at least partially, biotic-induced." (lines 777-779).

In the revised version: "Rain events appeared to attenuate the emission rates of VHOCs at the Dead Sea. Measurements at a bare soil site suggested a decrease in VHOC emission rates for 1−3 days after a rain event." (lines 935-937). We do not include the hypothesis about biotic-induced VHOC emission, because it is less strongly supported if we consider measured flux as positive or negative only for measurement sites identified as statistically significant ($p < 0.05$) sinks or sources for the specific species (for which the flux was measured).

4. In the original version: "Trihalomethanes, including $CHCl_3$, $CHBr_2Cl$, $CHBr_3$ and particularly $CHBrCl_2$, are associated with the highest number of sites at which their flux was, on average, positive, while $CHBr_3$, $CHBr_2Cl$ and $CHBrCl_2$ showed relatively high incidence of positive fluxes, with values of 65 %, 80 % and 85 %, respectively." (lines 780-783).

In the revised version we do not include this sentence as is, because this cannot be supported, if uncertainties in a measurement site as source are taken into account. In the revised version, we focus more on the common mechanisms/controls for the emission of brominated trihalomethanes: "Both flux and mixing ratio correlation analyses pointed to common formation and emission mechanisms for $CHBr_2Cl$ and $CHBrCl_2$, in line with previous studies, for the agricultural watermelon-cultivation field and bare soil sites. These analyses further strongly suggest common formation and emission mechanisms for $CHBr_3$ with these two trihalomethanes." (lines 938-942).

5. In the original version: "The overall average net flux of the trihalomethanes decreased according to $CHBr_2Cl > CHBr_3 > CHBrCl_2 > CHCl_3$." (lines 790-791).

In the revised version we further support this point by using the flux magnitude: "The overall average net flux of the trihalomethanes decreased according to $CHBr_2Cl > CHCl_3 > CHBr_3 > CHBrCl_2$, while $CHCl_3$ showed the lowest incidence of positive fluxes among all trihalomethanes." (lines 944-946). Again, this finding of relatively elevated emission of brominated trihalomethanes (compared to previous studies) is generally supported by both the original and revised analyses.

6. In the original version: "We identified the SD sites as a probable source for all methyl halides, whereas vegetated sites appear more likely to act as a net sink for these species." (lines 796-797).

In the revised version: we realize that this sentence should be revised based on both the original and updated analyses: "We identified the coastal sites as a probable source for all methyl halides, whereas neither agricultural field nor natural vegetation site were identified as net sink or net source for these species, except for the agricultural field being a net sink for $CH_3I$." (lines 949-951)

7. In the original version: "Comparing the proportion of Br and Cl in the soil for the various sites with proportions of measured positive flux of $CH_3Br$ and $CH_3Cl$ are in line with reports by Keppler et al. (2001) about emission of methyl halides via abiotic oxidation of organic matter in the soil. Similar calculations in our study demonstrated much higher efficiencies of $CH_3I$ emission than those reported by Keppler et al. (2000), pointing to emission of $CH_3I$ via other mechanisms. The high correlation of

CH$_3$I emission with that of CHCl$_3$ and C$_2$HCl$_3$, particularly at the SD sites, together with findings by Weissflog et al. (2005), of various chlorinated VHOCs emission, including CHCl$_3$ and C$_2$HCl$_3$, from salt lake sediments, suggests that the Dead Sea, particularly the SD, sites probably act as an emission source for CHCl$_3$, C$_2$HCl$_3$ and CH$_3$I via similar mechanisms. Weissflog et al. (2005) reported that the emission of chlorinated VHOCs in their study was induced by microbial activity. Keppler et al. (2000) reported the involvement of an abiotic process in the formation of alkyl from soil and sediments, and the observed correlation between methyl halides and both CHCl$_3$ and C$_2$HCl$_3$ may indicate that the two processes occur simultaneously." (lines 797-810).

In the revised version: The flux- and concentration-based correlation analyses in the revised version strongly support these findings, even though the coastal sites were not identified as statistically significant net sources for CHC$_3$l ($0.05 > p < 0.1$; see Table 2). The revised text refers to the statistical significance of the analyses and we have removed some experimental details about the related analyses to shorten the discussion: "Our analysis demonstrated, however, much higher efficiencies of CH$_3$I emission than of CH$_3$Br and CH$_3$Cl emissions as a function of halides in the soil, compared to those reported by Keppler et al. (2000), pointing to emission of CH$_3$I via other mechanisms. The strong correlation between both fluxes and mixing ratios of CH$_3$I, CHCl$_3$ and C$_2$HCl$_3$, particularly at the SD sites, strongly suggests that the coastal area of the Dead Sea acts as an emission source for CHCl$_3$, C$_2$HCl$_3$ and CH$_3$I via similar mechanisms, although these sites were associated with only moderate statistical significance ($p \leq 0.1$) as a net source for CHCl$_3$. The emission of CHCl$_3$ and C$_2$HCl$_3$ from these sites is in line with findings by Weissflog et al. (2005) of emission of various chlorinated VHOCs, including CHCl$_3$ and C$_2$HCl$_3$, from salt lake sediments. Weissflog et al. (2005) reported that the emission of chlorinated VHOCs in their study was induced by microbial activity. Keppler et al. (2000) reported the involvement of an abiotic process in the formation of alkyl from soil and sediments, and the observed correlation between methyl halides and between CH$_3$I and both CHCl$_3$ and C$_2$HCl$_3$ may indicate that the two processes occur simultaneously in the coastal area of the Dead Sea." (lines 951-965).

**4.** *Inferences about flux from measurements as a function of height have a certain spatial influence function. Please indicate what that might be for the sampling heights you have chosen.*
**Answer**: All measurement sites were carefully selected to ensure a sufficiently large homogeneous fetch, and the measurement height was chosen to ensure that the footprint falls within this homogeneous fetch, except for the SD sites where direct emission and uptake from the seawater can potentially affect the samples (lines 236-240). For EGD-SD-s and EGD-SD-w, the footprint included the seawater:" Based on the wind direction, in both cases, the sampling footprint included both the seawater and a narrow strip of bare soil mixed with salty beds (estimated at about 60% of the footprint) very close to the seawater." (lines 667-669). According to our calculations, the 80% footprint in the studied area ranged from ~100–950 m, which was, in all cases, significantly smaller than the fetch of any site (see Table S4). In some cases, the 90% footprint was ~2 km, but taking into account the wind direction for these specific cases (Table S6), the footprint was still smaller than the fetch, except for the samplings at the COAST–EGD, as described above. Based on this and the next comment, we realized that information was missing in the text about footprint and measurement height selections. Therefore, we have added the following text: "By default, the differences in height between the canisters increased exponentially with height, considering the typical decrease in the vertical gradient of emitted species in the surface layer (Stull 1988). All canisters were placed high enough above the ground to ensure that all sampling was performed within the inertial sublayer, except for the lowest canister at TMRX−ET. In all cases, the sample footprint fell inside the target fetch, except for the sampling at COAST–EGD, for which the sample footprint included a narrow strip of the seawater (estimated at about 40% of the footprint)." (lines 233-240).

**5.** *Consideration of the C2HCl3 results (an implied sink, perhaps from elevated mixing ratios in the broader Red Sea region) may indicate that the fluxes you are deriving here for naturally-emitted gases are actually not representative of the local regions you intended them to represent. How is the reader to assess this? Also, what has determined the different heights at which samples were collected on these masts? Sampling heights in a region with local emissions should have a large, but not discussed, impact on measured mixing ratios–which are being compared among sites and to MBL results.*

Answer: Our analyses suggest that in general, the investigated sites act more like a sink for $C_2HCl_3$, with the coastal sites probably also being a source for this species, based on the mixing ratios and flux correlation analyses (Tables 4 and 5), and the measured positive fluxes from these sites (Table 2). We cannot rule out that the sink for this species reflects its emission from the Red Sea, the Mediterranean Sea, or an anthropogenic source upwind. However, it is not likely that emission from the Red or Mediterranean Sea impacts the measured fluxes at the Dead Sea due to the following: (i) we added correlation and wind direction analyses, including for $C_2Cl_4$, which strongly support the origin of $C_2HCl_3$ from an anthropogenic source (see Sect. S4); (ii) the Red Sea and the Mediterranean Sea are located 160 km and 90 km from the Dead Sea, respectively, while mixing ratios of the investigated VHOCs at the Dead Sea are typically significantly higher than those in the MBL, and therefore probably also compared to those over the Red Sea and Mediterranean Sea; (iii) prevailing wind direction during the (different) measurement periods was from the north and in only a few cases, from the northwest (Table S6), whereas the Red Sea and Mediterranean Sea are located to the south and west of the Dead Sea, respectively; (iv) there is no reason to assume that the Dead Sea is an efficient sink for VHOCs transported from the Red Sea and the Mediterranean Sea, whereas there are efficient sinks for these species along the trajectories of the air masses. Therefore, it is not likely that the Dead Sea acts as a significant sink for VHOCs which are transported from these seas.

The Dead Sea probably acts more as a sink than a source for $C_2HCl_3$, but based on the above, this is more likely be the result of emission from inland anthropogenic sources in Israel. In any case, deposition of $C_2HCl_3$ or any other species cannot contribute to the emission fluxes at the Dead Sea—the latter, and not deposition, being the focus of this study.

Pursuant to this comment, we include the following: "Only COAST–EGD and COAST-TKM-SD sites were found to be statistically significant sources ($p < 0.05$, see Table 2) for $C_2HCl_3$, suggesting that the elevated mixing ratios for this species in the Dead Sea area result mostly from local anthropogenic emissions. This possibility is supported by the high correlations with $C_2Cl_4$ (Table S5). Emissions from a more distant natural source, such as the Mediterranean Sea or Red Sea, are unlikely given their large distance away (~90 km and ~160 km, respectively)." (lines 488-493).

To address the comment about sampling heights, we include the following: "By default, the differences in height between the canisters increased exponentially with height, considering the typical decrease in the vertical gradient of emitted species in the surface layer (Stull 1988). All canisters were placed high enough above the ground to ensure that all sampling was performed within the inertial sublayer, except for the lowest canister at TMRX−ET. In all cases, the sample footprint fell inside the target fetch, except for the sampling at COAST–EGD, for which the sample footprint included a narrow strip of the seawater (estimated at about 40% of the footprint)." (lines 233-240).

In the revised version, we discuss the impact of measurement height as well as of additional factors, including season and latitude, on differences in mixing ratios between our study and the MBL: "It should be noted, however, that while Fig. 2 implies elevated VHOC emission from the Dead Sea, comparison of mean or median mixing ratios of VHOCs for the Dead Sea with those for the MBL is not straightforward, considering that VHOC mixing ratios in the MBL are sensitive to several factors, including season and latitude. Moreover, the measurement height can play a significant role in affecting the mixing ratios due to decreasing mixing ratios with height over areas where local emissions occur. Hence, we also compared the measured fluxes and mixing ratios with their corresponding values measured in coastal areas, where the highest mixing ratios in the MBL were generally measured due to stronger emissions." (lines 347-355).

**6.** *I find the results in Figure 3 intriguing, although not much is made of it in the text. While it may be that no generalizations are possible related to all gases, there are some interesting similarities that might be worth discussing, especially to understand if these co-variations are consistent with the discussions related to co-variations in fluxes as what was intended in Table 4.*

**Answer**: First, note that Fig. 5 (originally Fig. 3) has been revised according to a comment made by reviewer #2 (we removed the data for VHOCs that did not show any seasonal variation), and the original figure is presented as Fig. S1 in the Supplement. The only case for which we suspect that there was a clear seasonal effect on VHOC mixing ratios is $CH_3I$, where both flux and mixing ratio measurements clearly indicated higher emission in spring vs. winter, apparently in line with findings by Sive et al. (2007), and accordingly we have added the following: "While no clear impact of season on mixing ratios was observed, for most sites, differences between two measurement sets resulted in consistent differences in mixing ratios, such that one measurement set resulted in higher mixing ratios for all or most species than the other. This suggests that other factors play a significant role in emission rates of all or most VHOCs in the studied area. Only the $CH_3I$ results indicated moderate statistical significance ($0.05 < p < 0.1$) for higher mixing ratios in the spring vs. winter, in agreement with seasonal trends for its flux, as discussed above." (lines 598-604).

We further add, in Sect. 3.2.3 on lines 763-766:" As discussed in Sect. 3.2.1, the mixing ratios of $CH_3I$ also tended to be higher in magnitude in spring compared to winter, with moderate statistical significance ($0.05 < p < 0.1$ in both cases) (Figs. 3, 5)."

Considering this comment, we have modified Table 4 (originally Table 5), and the correlations between mixing ratios are now provided individually for different site types, similar to Table 5 (originally Table 6), and also for all site types except for SEA–KDM (this site explores the effect of air transported over the seawater on the mixing ratios). These analyses indeed enabled us to further strongly support apparent common emission sources and/or controls between brominated trihalomethanes (Sect. 3.3, lines 846-882) and between $CHCl_3$, $CH_2Cl_3$ and $CH_3I$ (Sect. 3.3, lines 883-895), as well as between methyl halides (Sect. 3.3, lines 883-886).

**7.** *Table 4 is also very hard to extract information from... and as before, I'm concerned that any identification of positive flux amounts don't take into account uncertainties on those estimations. If uncertainties were not considered, then it seems that much of the discussion related to incidences of positive flux and rankings by chemical etc. that follows should be reconsidered.*

**Answer**:

We have realized that the information on the ratios between flux and mixing ratio (defined as F:C in the original version) does not contribute significantly to the manuscript, and based on this comment, we have removed the F:C information from the table (now Table 2). We believe that this makes the table easier to extract data from.

As explained above, we include in the revised manuscript *p*-values, indicating the statistical significance of the related measurement site being a source for a particular species. Only *p*-values $\leq 0.05$ were considered statistically significant in the analyses in the revised manuscript. The resulting differences in our conclusions and findings are summarized in our response to comment #3. Overall, they were not significantly changed by taking into account statistical significance.

**8.** *Table 5 and 6 need a consideration of correlations that are and are not significant, given the number of measurements included in each determination. Given the small number of samples considered here, I'd estimate that correlations of <0.1 are in fact indicative of no evidence for a correlation, not a correlation described as "low".*

**Answer**: We agree. All correlations in Tables 4 and 5 (originally Tables 5 and 6) are now reported with a corresponding *p*-value, based on Student's t-test, to indicate whether the correlation is significantly different from zero. Similar to the revisions we made to Table 2, we consider four different categories of *p*-values: $p < 0.05$, $p < 0.1$, $p < 0.15$ and $p > 0.15$. Only *p*-values $<0.05$ are considered statistically significant in the analyses in the revised manuscript.

**Response to comments by reviewer #2**

*General Comments*

*The manuscript by Shechner et al. presents ambient measurements and fluxes for short-lived halocarbons at multiple sites around the Dead Sea. The unique characteristics of the Dead Sea make it a very interesting location to study the emissions from and detail the characteristics of this source for atmospherically important halocarbons. The paper contains an abundance of information, but I feel some key details are lacking that are needed to fully assess the author's interpretations. Additionally, the paper is very long and becomes difficult to follow in*

*terms of the main points trying to be conveyed in the various sections of the paper. My opinion is that the paper could be distilled down in length and the key points be fleshed out a bit more cleanly. Additionally, I feel there are some significant improvements that could be made in dissemination of the information in both graphical and tabular form. While there is merit to the manuscript, I feel as though there are an array of issues that should be addressed before it is in an acceptable format for publication.*

*I will present a general list of issues here and elaborate on them in the Specific Comments section.*

*Urban and other source influences – it would be useful to provide some context to the potential of urban emissions, for the solvents like CHCl3 and C2HCl3, but also including things like wastewater treatment facilities and other agricultural activities that could influence the area.*

*The first time a chemical constituent is introduced, it should be spelled out – there are several places this occurs throughout the manuscript. For example, L72 chloroform (CHCl3), L73 chloroethane (C2H5Cl), L112 iron (Fe), L115 potassium bromide (KBr), nitric acid (HNO3), etc. – please address.*

*Percentages – there are spaces between the number and the percent sign. The most common convention is to not have a space between a number and the percent sign.*

*I would recommend either referring to the suite of halocarbons as VHOCs or VSLS, but not going back and forth between them.*

*From the measured fluxes, can you estimate the local/regional source or sink strength of the Dead Sea? How do your results play in to the scale of the source strength of the Dead Sea for these gases?*

*It would be useful to present some quantitative information in the abstract, such as mixing ratios and fluxes.*

*There are no uncertainties propagated through any of the fluxes.*
*I would be useful to include the atmospheric lifetimes and primary removal sources for the compounds in the manuscript.*

*The manuscript seem to try and agree with all previous studies.*
*Tables are difficult to read and digest.*
*Plots within the figures are too small making it difficult to extract information from them.*

*Flux section could be moved to SI*

*A more thorough overview of the site, including meteorology, would be useful to help set the stage for the reader.*

**Answer**: We have addressed all of the above according to the specific comments below and have revised the discussion to make it more concise. We have also added information in Sect. 2.1.1 regarding anthropogenic emission sources in the studied area, including agricultural fields, and mention that to the best of our knowledge, there are no wastewater facilities in the area of the Dead Sea. The reader is further referred to a sensitivity analysis which investigates, as suggested below, the potential impact of anthropogenic activity on the measured mixing ratios, based on the ratio $[C_2HCl_3]/[C_2Cl_4]$ (see Sect. S4). We further include a description of the meteorological conditions in the area of the Dead Sea (Sect. 2.1.1), and a summary of the meteorological measurements in Sect. S6.

In the revised version, we rigorously take into account the statistical uncertainties associated with the measured fluxes (and correlations) for our discussion and analyses, allowing us to present a clearer description of net emission/sink for the different species at the various investigated sites. Nevertheless, we do not yet have sufficient measurements to provide a reliable estimate of the overall emission/sink status for the various species at the Dead Sea. The Dead Sea area includes relatively highly diverse and changing landscapes, partly due to rapid evaporation, which leads to exposure of new deposits, and our measurements indicate high variability in emission, even for the same landscape. Therefore, many more measurements are required for a reliable estimate of total emission/deposition of VHOCs in this area. We have added flux and mixing ratio ranges for all species to the abstract and give lifetimes and primary removal pathways in Sect. S1, with a reference in the Introduction (line 84). We have revised original Figs. 2 and 3 and Table 2 for easier extraction of information, as suggested in the reviewer's comment below, and have included a new figure which provides a schematic of the flux's spatial distribution in the studied area.

***Specific Comments***
*L46-7: You should include why CH3I and C2HCl3 are exceptions, as this is not intuitive to the reader.*
**Answer**: The text in this case has been modified to account for statistical uncertainties associated with a measurement site being a source for the tested VHOC: "Fluxes were generally positive (emission into the atmosphere), corresponding to elevated mixing ratios, but were highly variable… Taking into account statistical uncertainties, the coastal sites (particularly those where soil is mixed with salt deposits) were identified as the source for all VHOCs, but this was not statistically significant for $CHCl_3$. Further away from the coastal area, the bare soil sites were sources for $CHBrCl_2$, $CHBr_2Cl$, $CHCl_3$, and probably also for $CH_2Br_2$ and $CH_3I$, and the agricultural sites were sources for $CHBr_3$, $CHBr_2Cl$ and $CHBrCl_2$." (lines 45-58).

*L49-51: For the statement: "Correlation analysis, in agreement with recent studies, indicated common controls for the formation and emission of all the above trihalomethanes but also for CH2Br2.", I'm not convinced this is entirely accurate – for example, what about CHCl3? Also how does the correlation indicate that the factors controlling the formation and emissions are the same?*
**Answer**: Two sentences have been included to address these two points based on the revised analyses which take into account statistical uncertainties: "Correlation analysis, in agreement with recent studies, indicated common controls for the emission of $CHBr_2Cl$ and $CHBrCl_2$, and likely also for $CHBr_3$. There were no indications for correlation of the brominated trihalomethanes with $CHCl_3$." (lines 61-

64). We agree that the correlation is not necessarily indicative of similar formation controls, but we believe that it is indicative of similar emission controls.

*L55: "elevate" should be "elevated"*
**Answer**: Corrected (line 68)

*L61: When you introduce VSLSs here, you should include here that this refers to compounds that have lifetimes of less than 6 months.*
**Answer**: This is now included (line 75).

*L64: replace "destruction of ozone" with "ozone destruction"*
**Answer**: Changed accordingly (line 78)

*L73: add "which", so it reads "…C2H5Cl, which originate…"*
**Answer**: Amended (lines 87-88)

*L134-5: Bromide (Br-) and chloride (Cl-) should be introduced and the sentence should be revised to read " with water salinity 12 times higher and a bromide to chloride ratio (Br−/Cl−) 7.5 times higher than in normal ocean waters.*
**Answer**: Amended (lines 151-153)

*L136: What do you mean by "landforms"? Formations from the residual salts left behind? In this case the use of the term "landform" invokes images of large scale topographical features, is this the case?*
*This brings in to question the use of the term landform in the title – is this really appropriate and accurate? I would say this work has been carried out on different terrains or ecosystems of the Dead Sea, but not different landforms.*
**Answer**: We have replaced the term landform with landscape (including the title) to indicate that the entire ecosystem is being addressed, and not just geological formations. It is true that some of the measurement sites represent similar landscapes, as in the case of different distances from the seawater in the same area (e.g., for COAST–TKM-SD, COAST–TKM-LD), but in most cases, they differ fundamentally in some aspect (e.g., soil mixed with deposits at the coast vs. bare soil far from the seawater vs. cultivated vegetation vs. natural vegetation) and the areas with the same characteristics are quite large (at least one to a few kilometers). We think that using "terrains" may imply different slopes, structures, etc. We replaced "landforms" on (original) line 136 with "newly exposed sea deposits" (line 154) to better describe the location.

*L143: I would revise this to make it a stronger statement, something like: "Studying the emission of VHOCs at the Dead Sea is also fundamental for understanding local surface ozone depletion events…"*
**Answer**: Thank you. We have modified the sentence accordingly (lines 161-164).

*L169-70: Regarding the Tamarix vegetation and watermelon fields, more details, such as density, proximity, size of agricultural development, etc., would be useful to the reader.*

*Also, I would refer change your referencing of watermelon fields from "vegetation" to "agriculture" in later sections of the manuscript – because this is a perturbed system different that the natural vegetation, it should be distinguished as such.*

**Answer**: More information is now included for the two sites, such as size, height, vegetation cover fraction and distance from the sea. In addition, meteorological conditions and estimated footprints are now included in the Supplementary Information (Tables S6 and S4, respectively). We refer in the revised manuscript to the watermelon field as an agricultural vegetation or agricultural field throughout the text. Note, however, that for the analyses, we refer to the natural vegetation and agricultural vegetation both as separate sites and as two vegetated sites (VEG; e.g., Tables 5 and 6).

*L198: Revise to "Lastly, WM-KLY…"*
**Answer**: Done (line 224)

***P8, Sect. 2.1.2.*** *How many samples were collected in total, at each site, and at each corresponding height for each site? What were the meteorological conditions during the sampling?*
*In order to get better feel for the results presented, both for the ambient levels and the fluxes, knowing N is critically important. This will allow the reader better perspective on some of the interpretation presented.*
*Also, general information about the seasonal and local meteorology to provide an overview of the region would be instructive to the reader.*
**Answer**: In the revised version, sample information is specified in Table 1 and the number of samples at each site is presented in Fig. 2 (replacing original Table 2), while Table 2 (originally Table 4) and Fig. 3 present each evaluated flux individually; we specify that in two cases, flux was evaluated based on two samplings rather than three. Tables 4 and 5 (which present the correlation analyses; originally Tables 5 and 6) present the number of values used for the mixing ratio and flux correlations, respectively. We have added to the Supplement a table that summarizes the collected meteorological data during the different measurements (Table S6) and refer to it in the text (line 246). In the revised version, we include information about the climate of the Dead Sea, including evaporation rate, annual precipitation and seasonal variation in daily maximum temperature (lines 177-181).

*L209: Regarding the use of "fast" here: I personally wouldn't consider 20 minutes to be fast - I think the key point you are trying to make is that all samples were collected simultaneously and integrated over a 20 min period – please revise.*
*Also, I'm assuming "lifting of the canisters" should be "filling of canisters"*
**Answer**: Thank you. The sentence has been revised to clarify this point: "To minimize non-synchronized air sampling by the three canisters, we constructed a special sampling system that allows almost simultaneous filling of the canisters" (lines 240-241).

*L209-12: Please revise the following sentence – very awkward as written:*

*Facilitated by passive grab samplers (RESTEK Corporation, PA, U.S.), we performed each sampling within 20 minutes by pulling air into evacuated 1.9 L stainless steel canisters, resulting in an internal canister pressure higher than 600 torr.*

**Answer**: The sentence has been revised: "For each sample, air was drawn into a 1.9 L stainless-steel canister via passive grab samplers (Restek Corporation, PA, USA), resulting in a sampling duration of 20 min and internal canister pressures higher than 600 Torr." (lines 242-244).

*L215-16: Please revise: "...subjected to the analytical techniques..." – simply say they were analyzed by similar techniques described in Colman et al.*

**Answer**: Amended (lines 246-248)

*L218-21: You introduce all of the halocarbons here, but most, if not all should have been introduced previously. Please address.*

**Answer**: Done

*L223-28: Please provide some statistical/quantitative rationale for this - you can't simply disregard this point because it doesn't "agree" with the other measured mixing ratios for CH3Cl. Also, I don't feel it's appropriate to state that it may result in a "less accurate flux" – how do we know what the "accurate flux" is? There is variability in all of this work, and while this may, in fact, be a spurious data point, what measures were carried out to deduce this issue?*
*Where is this listed, in Table 2? Please specify here for the reader to address.*

**Answer**: This is now supported using Grubbs's test (Grubbs and Beck, 1972) and the text was revised according the comment (Sect. 2.1.2, lines 252-255). We now indicate this issue in Table 1 as well, and refer the reader to Sect. 2.1.2 (where this issue is described) from all relevant figures and tables.

*L229, Table 1: It would be useful to provide the total number of samples and how many per height. This should be summarized such that the reader doesn't have to try and count how many samples were collected on the individual days from the information in the table.*

**Answer**: We now specify the total number of samples for each experiment in Table 1. Note that the number of samples corresponding with each of the correlation analyses is now specified in both Tables 4 and 5 (which present the correlation analyses; originally Tables 5 and Table 6) as well as Fig. 2 (replacing original Table 2).

*L279: Sect 2.3*
*Following suit with the canisters, how many total soil samples were collected and analyzed? This potentially could be moved to the SI because the information os only used for general properties at each site.*

**Answer**: We mention now in Table 3 that: "Analyses were performed for a single mixture of samples at each site." (lines 693-694). This was of particular importance for the watermelon field site, where the area was clearly not homogeneous. A different number of samples for each site was used for each sampled mixture. We prefer to keep the table in the main text because we use some of the information presented there in our discussion, mostly in Sect. 3.2.2, regarding the content of halides (e.g., lines 652-664) and organic matter (e.g., lines 769-780).

*L280-81: Please elaborate what you mean by this and what is the significance of this statement: "...at least 3 months following any rain event in the Dead Sea area."*

**Answer**: This is to ensure that sample composition and water content are not affected by drifts and percolation, following recent rain events in the area, as is now explained in the text:" Soil samples at each site were collected up to a depth of 5 cm during the summer, at least 3 months after any rain event in the Dead Sea area, to ensure no impact on the samples by recent drift and percolation." (lines 311-313).

*L265: In line reference should be Golder (1972)*

**Answer**: Corrected (line 297)

*L290: Quotes are not needed around "Discover"*

**Answer**: Done (line 322)

*L292-93: High Resolution does not need to be capitalized*

**Answer**: Done (line 324)

*L294: "low-limit" should be "lower limit"*

**Answer**: Amended (line 326)

*L302: What is meant by "corresponding available information."*

**Answer**: We have removed this sentence and refer first to the measured mixing ratios: "Overall, the measurements at the Dead Sea boundary layer revealed higher mixing ratios for all investigated VHOCs than their expected levels at the Mediterranean Sea and Red Sea MBL, indicating higher local emissions from the Dead Sea area" (line 333-335).

*L306: What is the "Dead Sea Works"?*

**Answer**: A short description of the Dead Sea Works has been added: "The main anthropogenic emission source in the area, apart from local transportation and a few small settlements, is the Dead Sea Works, a potash plant located to the south of most of the measurement sites (see Fig. 1)." (lines 184-186).

*L326: There were surface seawater, ambient air and direct flux measurements of $CHBr_3$ in Zhou et al., 2005 – how do these compare with the Dead Sea?*

**Answer**: Information about mixing ratios of $CHBr_3$ as well as fluxes for $CH_2Br_2$ and $CH_2Br_2$ based on Zhou et al. (2005) has been added on lines 428-434.

*L330: It appears that a range of values is missing after 2-60 pptv – there is simple "(-)"*

**Answer**: The range has been added (line 373).

*L335: Re C2HCl3 and CH3I - while the reader can look at the figure, it would be useful to also state in the text what these gases are doing, on average.*

**Answer**: This sentence has been deleted because it is not valid when statistical uncertainties are taken into account, which is the case in the revised manuscript. We now address these species as follows: "It can be seen that for all species, at least one of the six studied areas could be classified as a net source, with somewhat less sites being statistically significant net sources for $CHCl_3$, $C_2HCl_3$ and $CH_3I$. Note that as explained above, $C_2HCl_3$ was found to be affected by anthropogenic emission, which could explain the relatively less frequent identified emissions for this species "(lines 419-423). We also discuss the measured fluxes of $CH_3I$ throughout Sect. 3.1, and compare them to fluxes reported in the literature: "The positive fluxes measured at BARE−MSMR were similar to the measured soil-emission fluxes of $CH_3I$ reported by Sive et al. (2007) at Duke Forest, averaging ~0.27 nmol m$^{-2}$ d$^{-1}$ (range, ~ 0.11−4.1 nmol m$^{-2}$ d$^{-1}$)." (lines 485-487).

*L336-39: Can you please clarify these two sentences: Figure 2 doesn't show values higher than these. Either present the values or revise text.*

**Answer**: The sentence has been revised: "The flux magnitudes for $CHBr_3$ and $CH_2Br_2$ were greater than for most reported emissions in the MBL (e.g., $CHBr_3$, 25.2–62.88 nmol m$^{-2}$ d$^{-1}$ for the Mauritanian upwelling (Quack et al., 2007); $CH_2Br_2$, 0.14–0.29 nmol m$^{-2}$ d$^{-1}$ for the New Hampshire coast (Zhou et al., 2008)), but were smaller than the corresponding average fluxes estimated by Butler et al. (2007) for global coastal areas (~220 and 110 nmol m$^{-2}$ d$^{-1}$, respectively) and than the average flux from the New Hampshire coast as reported by Zhou et al. (2005) (~620 ± 1370 nmol m$^{-2}$ d$^{-1}$ and 113 ± 130 nmol m$^{-2}$ d$^{-1}$, respectively)." (lines 428-434).

*L376: for the following, you either have one too many or one too few brackets: (e.g., ~600 nmol m−2 d−1; (Deventer et al., 2018).*
**Answer**: Corrected (line 473)

*L360: For "nmol m−2−d-1" there appears to be an extra dash in between m-2 and d-1*
**Answer**: Corrected (line 456)

*L391-93: It is difficult to see this in Fig 2, and what/where are the anthropogenic emissions located? From DSW or other places? Can this be assessed by looking at something like the C2HCl3/C2Cl4 ratio? It is likely that this data is available from the UCI group, but this (or other pairs of compounds) could be used to do a more thorough analysis on the impact of anthropogenic emissions at the sampling sites. For example, this brings in to light things like wastewater treatment facilities and the corresponding emissions of CHCl3 and CHBr3. It would be useful to provide a more rigorous assessment of the influence of anthropogenic emissions in general - particularly for those not familiar with the region and to what extent they may be influencing this work - if minimal, that's great - just demonstrate this, as this statement affects your results - C2HCl3 isn't the only gas here with anthropogenic sources.*
**Answer**: We have added more details to the Methods about potential anthropogenic emission sources in the area (lines 184-190), and a thorough analysis of potential anthropogenic emission sources in general and particularly for $C_2HCl_3$ (Sect. S5). We summarize this analysis in the main text: "No association was observed between the measured mixing ratios and the air masses flowing from the direction of the Dead Sea Works (see Sect. S4 for anthropogenic impact), a potash plant located to the northwest of the TMRX−ET site and to the south of all other measurement sites (see Fig. 1) that is the main anthropogenic source in the area under investigation. Furthermore, the correlation analysis (Table S5) revealed that only $C_2HCl_3$ was associated with $C_2Cl_4$, a well-known anthropogenic VHOC. The absence of any other associations suggested dominance of natural sources for the VHOCs in the studied area." (lines 335-342). To the best of our knowledge, no wastewater facilities are located near the Dead Sea water.

The sentences have been revised: "Only COAST–EGD and COAST-TKM-SD sites were found to be statistically significant sources ($p < 0.05$, see Table 2) for $C_2HCl_3$, suggesting that the elevated mixing ratios for this species in the Dead Sea area result mostly from local anthropogenic emissions. This possibility is supported by the high correlations with $C_2Cl_4$ (Table S5)." (lines 488-491).

*Suggestion: After looking at Figure 2, I feel as though it would be useful to have a summary flux figure (e.g., by compound) with the magnitude of the fluxes plotted by size or color on a map to enable the reader to get a better idea of the spatial variability of the flux magnitudes.*

**Answer**:
Thank you. We have produced a new figure—Fig. 4—which focuses on the spatial distribution and variation in measured net fluxes at the various site types. The figure clearly demonstrates higher emission from the coastal area, particularly for the sites which are closer to the seawater and from the natural vegetation for some of the VHOCs, which are generally higher than for the cultivated field. However, the natural vegetation site could not be classified as a statistically significant source for the investigated VHOCs, pointing to the need for additional measurements at this site (lines 555-560). We also keep Fig. 2, which is focused on the effect of season and distance from the seawater.

*L418: Revise to: "…VHOCs, except C2HCl3, were…"*
**Answer**: The sentence has been revised: "The results presented in Sect. 3.1 showed elevated mixing ratios and net fluxes for all investigated VHOCs, with relatively less frequent positive fluxes for $CH_3I$, $CHCl_3$ and $C_2HCl_3$." (lines 547-548).

*L431-33: Regarding the statement that there isn't a difference between fluxes in the spring and winter, two things should be addressed: 1) is this statistically significant? 2) What is the seasonality of the temperature and overall meteorology for this area (i.e., local/regional transport patterns)? Being only slightly extratropical, would seasonality be expected to be an important driver?*
**Answer**: These two points have been addressed as follows: "Differences in VHOC emissions between winter and spring may arise from the generally much higher temperature, and lower precipitation during the latter; further considering the high evaporation rate in this area, the soil water content is expected to be generally lower in spring compared to winter (Sect. 2.1.1; see also Table S6). Figure 3 suggests that there were no clear differences in VHOC fluxes between spring and winter, as supported by statistical analysis, except for $CH_3I$ and $CH_2Br_2$ for which fluxes were higher in the spring, with moderate statistical significance ($0.05 < p < 0.1$)." (lines 578-584).

Seasonal variations between spring and winter in this area are relatively significant (e.g., summer mean daily maximal temperature of ~40 °C decreasing gradually until winter, to a corresponding temperature of ~21 °C; Sect. 2.1.1). Basically, the area is controlled by 19 main synoptic scenarios, with a dominant influence of local sea and Mediterranean breezes (Shafir and Alpert, 2011). In the manuscript, we focus directly on the meteorological parameters (which are further summarized in Table S6 for the measurement periods), and soil water content (which was not measured for the specific measurement periods), rather than the dynamics and synoptics that control these parameters.

*L437: add comma after "properties"*
**Answer**: Done (line 587)

*L439-40: Regarding the sentence: "No clear impact of season or distance from the seawater on the mixing ratios can be discerned in this figure,…", while I agree, it's mostly because you can't see the details in Figure 3.*
*Figure 3: In general, it is difficult to discern the spatial distributions and get useful information out of the vertical profiles because each panel is so small. From this figure, it is difficult to see and discern the gradients for many of the gases. I would recommend revising and either show a few key species and put the remainder that don't show anything in the SI or revise the whole figure.*
**Answer**: We have revised the figure as suggested (Fig. 5 in the revised manuscript), by excluding information for those species that did not show any seasonal trend. This indeed enables us to present a less "busy" figure with somewhat larger panels. We include the original figure in the Supplement (Sect. S2). Note that the aim of this figure is to explore the effects of season and distance from the seawater on the measured mixing ratios and therefore, we use a similar scale for the y-axis associated with the same species in the same coastal area (e.g., COAST–TKM-SD and COAST–TKM-LD). As a result, in many cases, it is still difficult to see or discern the gradients in mixing ratios due to the substantial range. Pursuant to this comment, we now indicate in the caption of Fig. 5 that the figure uses even y-axis scaling for sites in the same coastal area (line 615). To make it clear that the figure focuses on mixing ratio differences vs. distance, we have arranged the panels differently and schematically indicate the distance from the seawater by arrows.

*L460: replace "these parameters" with "the soil composition parameters"; also change "The table records…" to "The results presented in Table 3 show…"*
**Answer**: Amended (lines 627-628)

*L462: "larger distance" should be replaced with " greater distances"*
**Answer**: Amended (lines 629-630)

*L465-66: replace "in" with "at" just before the site location abbreviation.*
**Answer**: Done (lines 633)

*L472-75: What do you mean by "underestimated value of Fe"? A lower limit of the total iron? Again, "low-limit" should be "lower limit".*
**Answer**: Yes, we mean a lower limit of the total iron. The sentence has been changed accordingly (line 639).

*What does "while the emission rates became saturated" mean – I'm assuming that you mean "plateau". For example, Huber et al. use the term "plateau".*
**Answer**: Yes, the sentence has been clarified: "Note, however, that soil Fe content similar to that reported here as a low-limit value corresponds with those associated with the finding of small amounts of VHOC emissions, while the emission rates become saturated when enrichment with Fe(III) is relatively minor (Keppler et al.,

2000). Saturation at relatively low soil Fe concentrations was also reported by Huber et al. (2009)." (lines 640-643).

*L478: I would replace "merges" with "combines"*
**Answer**: Table 2 (originally Table 4) is now introduced in Sect. 3.1, to support the discussion of flux magnitude with the statistical analysis that is incorporated into the table. We have also removed the F:C parameter from the table (according to one of the following comments), such that we do not use "merges" in the revised sentence (line 408).

*L481: "samplings" should be "sample", and I would encourage revising this sentence to something like: "While the number of samples collected at each site was limited, Table 4 shows that the fluxes...."*
**Answer**: The sentence has been revised accordingly (lines 645-647; see response to next comment).

*L482: I would replace "In both" with "For the...sites,..."*
**Answer**: The sentence has been revised according to this and the previous comment: "While the number of samples collected at each site was limited, Table 2 and Fig. 4 indicate elevated positive fluxes for the SD sites, and to some extent also at COAST−EGD-MD, with respect to both statistically significant and non-statistically significant positive fluxes" (lines 645-647).

*L484: replace "in" with "at" before COAST-EGD-MD*
**Answer**: Corrected (line 646; see answer to previous comment)

*L485: A comma is needed after "winter"*
**Answer**: Corrected (line 648)

*L505: For consistency, replace VSLS with VHOC.*
**Answer**: This text has been deleted, because it is less strongly supported when the statistical uncertainties in our analyses are taken into account. We generally use VHOCs through the text instead VSLS, except for the Abstract, Introduction and twice in the Results and discussions, where we specifically refer to VSLS (lines 347 and 358).

*L506-07: Replace "during" with "at" before the site abbreviations.*
**Answer**: We have separated the sentence into two sentences and revised them: " COAST−EGD-SD-s was associated with the highest incidence of both statistically significant and non-significant positive fluxes. Fluxes at COAST−EGD-SD-w were generally lower and with a smaller incidence of positive fluxes." (lines 665-667).

*L515-19: Do you need the F:C ratio really aid in understanding these processes?*
**Answer**: The F:C ratio has been removed from the table, because we are also able to support our discussion without it, and because it makes the table easier to extract information from (see also response to the next comment).

*L544: Table 4. General comment: This is a hard table to read and extract information from - I almost feel as though presenting this graphically would be more impactful allowing the reader to see the trends rather than sifting through a lot of numbers that appear to vary greatly.*
*Because everything is bolded in the summary portion of Table 4, the rows should be explicitly labeled as to what the values are.*

**Answer**: We have removed F:C from the table and we believe that this makes it easier to extract data. Information in the summary section of the table has been changed (including information which relates to statistical significance, instead of the F:C information), and we also explicitly label the rows in the summary portion. Note that Table 2 (originally Table 4) has been moved to Sect. 3.1 to support the discussion of fluxes from the various sites with the statistical analysis, which is incorporated into the table. The table now appears on line 494.

*L553: Awkward as written, say something like: The results presented in Table 4 show that a higher…"*
**Answer**: Amended (lines 698-699)

*L554: comma is needed after CHCl3*
**Answer**: The sentence has been revised according to the updated analysis which takes into account the statistical uncertainties: "Differently than previous studies, brominated VHOCs had relatively higher overall incidence of positive fluxes than chlorinated VHOCs (Table 2)." (lines 698-699).

*L555: "tends" should be "tended"*
**Answer**: The sentence has been revised, also based on the updated analysis, which takes into account the statistical uncertainties of a site as a net source for the species (lines 699-702).

*L558: I would suggest deleting the following (not needed): "suggesting both high emission and their balance to some extent by sinks for this species."*
*Because there were watermelon fields, was there any harvesting or drying and decomposing plant material in the vicinity of the sampling? This can be a source of an array of halocarbons, particularly gases like CHCl3 and CHClBr2.*
**Answer:** The whole sentence has been deleted, because it is less strongly supported by the updated statistical analysis. There was no harvesting or drying during the measurement periods. Overall positive net fluxes from the agricultural field site were not elevated compared to the bare soil sites, and tended to be, in most cases, comparable to or lower than those from the natural vegetation (TMRX–ET) (see new Fig. 4).

*L563: I would replace "are in general" with "were"*
**Answer:** Amended (line 706)

*L566-70: Please revise – it is unclear what you are trying to say.*
   **Answer**:  We have revised the specified text for clarity: "The latter explanation may be supported by the fact that Albers et al. (2017) did not find any correlation between $CHCl_3$ emission rate and organic Cl in the soil. Furthermore, our study points to higher emission rates and incidence of VHOCs, and generally also of trihalomethanes, closer to the seawater (COAST−EGD and COAST−TKM sites), which suggests higher sensitivity to soil halide content than OM (Sect 3.2.2)." (lines 709-714).

*L577-9: revise to something like: "...emission rates from both bare and vegetated soil sites supports the work by Albers et al. (2017) concerning the emission of trihalomethanes from the soil after trihaloacetyl hydrolysis (Table 3)."*
**Answer**: Thank you. The text has been revised accordingly (lines 721-723). Note that the natural vegetation site is not statistically significantly a source for the trihalomethanes (Table 2).

*L584: Agricultural emissions, such as from the watermelon farming, could be such a source. More details regarding the scale and influence of these operations would be useful.*
**Answer**: While emissions of both $CHBr_3$ and $CHBr_2Cl$ were observed at the agricultural field, even higher emission rates for these species were observed at the coastal sites, and for $CHBr_2Cl$ also from bare soil (BARE–MSMR-2, BARE–MSD-2, BARE–MSD-3 and BARE–MSD-4; see Table 2 (originally Table 4) and new Fig. 4), where there is no agricultural activity. Based on this comment, we now include the following: "note that agriculture could potentially be a source for the emission of $CHBr_2Cl$ and $CHBr_3$ for WM–KLY, but not for the other sites (Sect. 2.1.1)." (lines 729-731). Further, referring to the flux correlations between $CHBrCl_2$, $CHBr_2Cl$ and $CHBr_3$, we mention in Sect. 3.3 that: "Note that these correlations can potentially be attributed to agricultural emission, considering that WM–KLY, but not TMRX–ET, was identified as a statistically significant source for the three trihalomethanes" (lines 848-850).

*L589: replace "in" with "at" before the site name*
**Answer**: Amended (line 737)

*L589-91: Please revise the following – awkward as written: "No clearly more elevated positive flux of brominated compared to chlorinated trihalomethanes was observed for this site…"*
**Answer**: The sentence has been rephrased (lines 737-740).

*L600: include (Table 4) to direct the reader to this information*
**Answer**: Done (line 748)

*L601: For the statement "...indicating strong emission and deposition…", if the flux is positive, then the emissions outweigh the deposition or other loss processes - revise to clarify your point. Figure 2 counters the point of "strong deposition" for the methyl halides.*
**Answer**: The sentence has been revised based on this comment and the updated analysis that takes into account statistical uncertainties: "A relatively high incidence of negative fluxes was observed for $CH_3Br$, and more statistically significantly so for $CH_3Cl$ and $CH_3I$, implying high rates of both emission and deposition, at least for the latter two, in the studied area (Table 2)." (lines 746-748).

*L604-05: Cultivated watermelon fields (agricultural emissions) are different from local vegetation, please distinguish as such.*

**Answer:** "local vegetation" has been replaced with "agricultural field" (line 753).

*L644-46: please revise, reads awkwardly*
**Answer:** The text has been revised: "It should be noted, however, that the fluxes that we used for the methyl halide emission efficiencies were based on measured net flux rather than measured emission flux. This might also explain the inconsistency between the relative $CH_3I$-emission efficiency calculated by Keppler et al. (2000) and by us" (lines 795-798).

*L662-663: How are the data grouped for Table 5? Is this simply for all sampling heights lumped together? Is there a difference when grouped by height?*
*Replace "evaluated" with "measured"*
**Answer:** Table 4 (originally Table 5) refers to lumped correlations, to avoid higher correlations due to systematic trends of mixing ratios with height. Where height was taken into account, correlations tended to be higher and more statistically significant, but this does not well represent the correlations between species. Pursuant to this comment, we have added the following to the caption of Table 4:" Correlations were calculated for mean mixing ratios at each site" (line 832). The sentence has been revised:" Table 4 presents the Pearson correlation coefficients (r) between the measured mixing ratios of VHOCs at the Dead Sea, separately for all sites and for the terrestrial sites only, as well as separately for BARE, COAST, and the natural vegetation and agricultural field sites (VEG). For COAST, r is also presented individually for the two sites which were closest to the seawater (SD)." (lines 813-817).

*L670: Please consider revising: "...reinforce predominant contribution of VHOCs from terrestrial sources…" – I would consider this to be an overstatement.*
**Answer:** The sentence has been revised: "Correlations were in most cases either similar or smaller when we included measurements from the seawater site SEA−KDM, which may reinforce the notion that emission from the seawater does not contribute significantly to VHOC mixing ratios in the area of the Dead Sea." (lines 825-828).

*L672: The r2 values are quite low, and without being able to see the correlation plots of these gases, it is difficult to adequately assess the commonality of their sources and sinks. How do these specific r2 values translate in to common sources and sinks?*
**Answer:** In the revised version, we indicate the *p*-value associated with the presented correlation values. We also present the correlations individually for the different site types. This enables us to better support the correlation between $CHCl_3$ and methyl halides, at least for some of the site types (Table 4). Because r values (we use Pearson correlation coefficient (r) rather than coefficient of determination ($r^2$) in the revised version) cannot be directly translated into common sources and sinks, the short discussion about correlations according to Table 4 is followed by a discussion based on the flux analysis (Table 2) and flux correlation analysis (Table 5). In particular for the correlation between $CHCl_3$ and the methyl halides, the related discussion is as follows: " Interestingly, in agreement with Table 4, Table 5 also shows relatively high correlations between $CHCl_3$ and all methyl halides, particularly for the BARE sites ($CH_3I$, r = 0.68, $p < 0.15$; $CH_3Br$, r = 0.83, $p < 0.05$; $CH_3Cl$, r < 0.86, $p < 0.05$), and SD sites ($CH_3I$, r = 0.99, $p < 0.05$; $CH_3Br$, r = 0.59, $p > 0.15$; $CH_3Cl$, r = 0.91, $p < 0.1$). Remarkably, a high correlation was found for $CH_3I$ with $CHCl_3$ and $C_2HCl_3$ at the SD sites (r = 0.99, $p < 0.05$ in both cases). Positive fluxes of the three species were observed at the SD sites in most cases, although with only moderate statistical significance for $CHCl_3$ (Table 2). Weissflog et al. (2005) found that emission of $C_2HCl_3$, $CHCl_3$ and other chlorinated VHOCs can occur from salt lakes via the activity of halobacteria in the presence of dissolved Fe (III) and crystallized NaCl. The strong correlations of $CHCl_3$, $C_2HCl_3$ and $CH_3I$ at the SD sites, where statistically significant fluxes were frequently measured for these species, reinforce the colocalized emission of $CHCl_3$ and $C_2HCl_3$ from salt lake sediments, as indicated by Weissflog et al. (2005), and suggest that $CH_3I$ can be emitted in a similar fashion. The fact that the relative emission efficiency of $CH_3I$ in our study was much higher than under the conditions used by Keppler et al. (2000) supports the possibility that mechanisms other than the abiotic emission pathway proposed by Keppler et al. (2000) influence the emission of $CH_3I$ at the Dead Sea (Sect. 3.2.3)." (lines 883-898). Hence, we believe that the integrated analyses indicate a common source for $CHCl_3$, $C_2HCl_3$ and $CH_3I$. Note that, considering this comment, we removed the original sentence from the manuscript.

*L678: Replace "records" with something like "shows"*
**Answer:** Amended (line 838)

*L680-81: Change "For the two last,…" to something like: "For the latter two sites,…"*
**Answer:** Amended (line 840)

*L685: Replace "demonstrates" with something like "shows" or The results in Table 6 show/illustrate…*
**Answer:** Amended (line 843)

*L692-94: Can you please expand upon the correlations being attributable to "common sinks" – what are the sinks and how is this driving the correlations?*
**Answer:** Taking into account statistical uncertainties in our revised analyses indicated that the cultivated and natural vegetation could not be classified as statistically significant as either sink or source for methyl halides. Therefore, considering also this comment and the length of the discussion in Sect. 3.3, we have removed this specific discussion.

*L740: replace "common emission" with something like "co-located emissions"*
**Answer:** Done (line 893)

*L826-27: I would recommend revising or omitting the following: "...from saline soil and salt lakes in stratospheric and tropospheric chemistry,…", as there were no linkages made to how the compounds measured for this work play in to the local/regional/global budgets of tropospheric or stratospheric Cl, Br or I.*

**Answer:** We agree, and the text has been revised accordingly: "Overall, along with other studies, the findings presented here highlight the potentially important role of saline soil and salt lakes in VHOC emission, and call for further research on VHOC emission rates and controlling mechanisms, and implications on stratospheric and tropospheric chemistry." (lines 974-977).

[revised manuscript text omitted]

~~Considering, however, the large distance of the Red Sea (~160km) and the Mediterranean Sea~~

~~(~90km) from the Dead Sea, it is unlikely that emission from these sources lead to significant~~

**Table 2.** VHOC flux. Shown are
the measured flux (nmol m$^{-2}$ d$^{-1}$) obtained for the different measurements. Values in bold and in parentheses
indicate that the related measurement site is a significant ($p < 0.05$) or non-significant ($p > 0.15$) net source or sink
for the specific VHOC based on one–sample t-test. Additional categories are defined below. These calculations
assume COAST–EGD-SD and COAST–TKM-SD as the same source (see Section 2.1.2). Also shown are the
average flux (mean) and average positive flux (mean positive) for all species, as well as the percentage of incidence
of positive flux ($X$) out of total measured fluxes, individually for each site and each VHOC (See Table 1 for
abbreviations of the different measurement sites). All presented values, including mean, mean positive and $X$
include only fluxes associated with $p < 0.05$ (bolded; S) and values associated with $p \geq 0.05$ (presented in
parentheses; NS), based on one–sample t-test.

| Species / Site | CH$_2$Br$_2$ | CHBr$_3$ | CHBr$_2$Cl | CHBrCl$_2$ | CHCl$_3$ | C$_2$HCl$_3$ | CH$_3$Cl | CH$_3$Br | CH$_3$I | X (%) |
|---|---|---|---|---|---|---|---|---|---|---|
| BARE−MSMR-1 | **1.43** | (-76.5) | **-3.27** | **7.68** | **247** | (7.33) | (2629) | 71.9[b]9[a] | **4.42** | **33** (78) |
| BARE−MSMR-2 | **1.51** | (27.6) | **21.3** | **19.9** | **6.51** | (-10.4) | (-378) | 12.6[b]6[a] | **1.00** | **44** (78) |
| BARE−MSD-1 | (-55.4) | (-37.7) | **-3.58** | **1.32** | (12.1) | -11.0[e]0[b] | **-1266** | (5.26) | **-0.73** | **11** (33) |
| BARE−MSD-2 | (23.5) | (103) | **41.8** | **24.5** | (-6.02) | -24.8[e]8[b] | **-1368** | (-50.3) | **-8.14** | **22** (44) |
| BARE−MSD-3 | (-0.60) | (32) | **8.69** | **7.92** | (-14.6) | 4.32[e]32[b] | **311** | (-47.9) | **-2.95** | **22** (56) |
| BARE−MSD-4 | (-4.61) | (-1.41) | **26.96** | **19.1** | (64.7) | 6.39[e]39[b] | **-472** | (38.44) | **-3.58** | **22** (56) |
| COAST−EGD-SD-w | **0.85** | **78.1** | **90.0** | **6.63** | (-42.8) | **47.3** | **-1040** | **88.4** | **1.45** | **78** (78) |
| COAST−EGD-MD-w | **-6.53** | **-79.0** | **187** | **23.1** | (38.5) | **37.5** | (9719) | **-111** | -5.16[b]16[a] | **33** (56) |
| COAST−EGD-LD-w | **-16.7** | **88.7** | **768** | **-14.2** | (-43.7) | **-8.97** | (-2281) | **116** | -24.5[b]5[a] | **33** (33) |
| COAST−EGD-SD-s | **3.71** | **187** | **72.3** | **14.8** | 883[b] | 884[b]884[a] | **10817** | **118** | **17.0** | **78** (100) |
| COAST−EGD-MD-s | **1.35** | **48.6** | **13.4** | **3.42** | 46.4[b]4[a] | -8.39[e]39[b] | (-530) | **8.10** | **2.27** | **67** (78) |
| COAST−EGD-LD-s | **2.52** | **66.0** | **13.8** | **8.68** | -40.8[b]8[a] | -2.03[e]03[b] | 261[b]261[a] | **22.3** | **-2.96** | **56** (67) |
| COAST−TKM-SD-w | -4.15[e]15[b] | **-28.1** | **123** | **1.62** | 22.8[b]8[a] | **0.89** | **4895** | **110** | **2.42** | **67** (78) |
| COAST−TKM-LD-w | **2.95** | (28.5) | (-408) | (-6.2) | (-32.9) | -22.0[e]0[b] | **2200** | **57.3** | (-1.03) | **33** (44) |
| COAST−TKM-SD-s | 3.80[e]80[b] | **87.7** | **42.7** | **21.4** | 0.99[b]99[a] | **2.00** | **1210** | **49.3** | **-0.38** | **67** (89) |
| COAST−TKM-LD-s* | **0.56** | (-3.83) | 2.07[b]07[a] | (1.67) | (12.6) | -0.31[e]31[b] | **1100** | **23.6** | (0.97) | **33** (78) |
| TMRX−ET-1** | (-8.93) | (-23.0) | (-8.64) | (-28.5) | 27.6[b]6[a] | -0.36[e]36[b] | (10500*) | (-90.8) | (-6.14) | **0** (11) |
| TMRX−ET-2 | (70.6) | (73.7) | (20.4) | (45.4) | 213[b]213[a] | -4.53[e]53[b] | (-5300) | (10.9) | (3.61) | **0** (78) |
| WM−KLY-1 | 1.45[b]45[a] | **50.7** | **2.09** | **8.57** | (-577) | -74.1[e]1[b] | (983) | **53.5** | **-4.01** | **33** (56) |
| WM−KLY-2 | 11.3[b]3[a] | **24.5** | **12.6** | **8.76** | (6.31) | -20.0[e]0[b] | (-4730) | (-31.6) | **-8.29** | **33** (67) |
| | | | | | | | | | | |
| Mean — S | **-0.84** | **52.4** | **88.5** | **10.2** | **70.9** | **-2.2** | **1640** | **48.2** | **-2.75** | |
| Mean — NS | (1.43) | (32.3) | (51.1) | (8.78) | (41.2) | (40.1) | (1360) | (22.7) | (-1.74) | |
| Mean positive — S | **1.86** | **78.9** | **102** | **11.8** | **127** | **21.9** | **3400** | **65.9** | **6.17** | |
| Mean positive — NS | (9.66) | (68.9) | (90.4) | (13.2) | (122) | (124) | (4060) | (52.4) | (4.14) | |
| X (%) — S | **40** | **40** | **70** | **75** | **10** | **20** | **30** | **45** | **20** | |
| X (%) — NS | (65) | (65) | (80) | (85) | (65) | (40) | (55 ) | (75) | (40) | |

 a 0.05 < *p* < 0.1 for a measurement site as net source or sink for a specific species.

b 0.1 < *p* < 0.15 for a measurement site as a net source or sink for a specific species; S and NS indicate *p* < 0.05 and

*p* > 0.05, respectively.

* Flux calculation excludes one measurement for all VHOCs (see Sect. 2.1.2).

** Flux calculation excludes one CH₃Cl sample (see Sect. 2.1.2).

[Figure]

**Fig.ure 3.** VHOCs fluxes at the different measurement sites. Fluxes associated with $p$-values <0.05 are marked by colored circles to indicate measurements during spring, winter, and summer, with full- colored circles indicating measurements up to 3 days after a rain event in spring (Spring-prec.), up to 6 days after a rain event in winter (Winter-prec.) and in the evening in summer (Summer-eve). Gray and black shapes indicate fluxes associated with no clear statistical significance ($p > 0.1$ and $0.05 < p < 0.1$, respectively). At the center of each graph, the small black circles and error bars represent the average and standard error of the mean (SEM), respectively, for each measurement site. Dashed lines represent zero flux. In each box, the numbers indicate the mean flux and SEM (in parentheses) for each site and species. Additional information is provided about measurement conditions (Tables 1, and S6), measurement abbreviations (Table 1) and statistical analysis (Table 2). *Calculation of $CH_3Cl$ flux mean and SEM excludes one sample at TMRX−ET-1 (see Sect. 2.1.2). **Calculation of mean flux and SEM excludes one sampling canister at COAST–TKM-LD (see Sect. 2.1.2).

**3.2 Factors controlling  VHOC flux **

**3.2.1 Seasonal, meteorological and spatial effects**

The results presented in Sect. 3.1  showed elevated mixing ratios and net fluxes for all investigated VHOCs, with relatively less frequent positive fluxes for $CH_3I$, $CHCl_3$ and $C_2HCl_3$.

For all of the investigated VHOCs, a  positive flux was measured for at least one of the two bare soil sites,

BARE−MSMR and BARE−MSD, which are located a few kilometers from the Dead Sea water.

For several VHOCs ($CH_2Br_2$, $CHBr_2Cl$, $CHBrCl_2$ and $CHCl_3$), at least one of these sites was identified as a significant net source ($p < 0.05$, Table 2). Additional measurements are required to determine whether the other VHOCs are also emitted from these bare soil sites. Note that̶a̶n̶d̶ for all V̶O̶HOCs̶,̶ except $C_2HCl_3$ and $CH_3Cl$, ̶a̶l̶l̶ measured mixing ratios were highest over at least one of these bare soil sites (Table S3̶2̶). ̶T̶h̶e̶s̶e̶ ̶f̶i̶n̶d̶i̶n̶g̶s̶ ̶s̶u̶g̶g̶e̶s̶t̶ ̶t̶h̶a̶t̶ ̶a̶ ̶s̶i̶g̶n̶i̶f̶i̶c̶a̶n̶t̶

̶e̶m̶i̶s̶s̶i̶o̶n̶ ̶f̶o̶r̶ ̶a̶l̶l̶ ̶o̶f̶ ̶t̶h̶e̶ ̶i̶n̶v̶e̶s̶t̶i̶g̶a̶t̶e̶d̶ ̶V̶H̶O̶C̶s̶ ̶o̶c̶c̶u̶r̶r̶e̶d̶ ̶f̶r̶o̶m̶ ̶b̶a̶r̶e̶ ̶s̶o̶i̶l̶ ̶l̶o̶c̶a̶t̶e̶d̶ ̶w̶i̶t̶h̶i̶n̶ ̶a̶t̶ ̶l̶e̶a̶s̶t̶ ̶a̶ ̶f̶e̶w̶

[revised manuscript text omitted]